# Poldip2 promotes mtDNA elimination during *Drosophila* spermatogenesis to ensure maternal inheritance

Ziming Wang[1,2,8], Tirawit Meerod[3,8], Nuria Cortes-Silva[1,2,5,8], Ason C-Y Chiang[3,8], Ziyan Nie[1,2,6], Ying Di [1,2,7], Peiqiang Mu [4], Ankit Verma [1], Adam James Reid[1] & Hansong Ma [3✉]

## Abstract

**Maternal inheritance of mitochondrial DNA (mtDNA) is highly conserved in metazoans. While many species eliminate paternal mtDNA during late sperm development to foster maternal inheritance, the regulatory mechanisms governing this process remain elusive. Through a forward genetic screen in *Drosophila*, we identified 47 mutant lines exhibiting substantial retention of mtDNA in mature sperm. We mapped one line to *poldip2*, a gene predominantly expressed in the testis. Disruption of *poldip2* led to substantial mtDNA retention in mature sperm and subsequent paternal transmission to progeny. Further investigation via imaging, biochemical analyses and ChIP assays revealed that Poldip2 is a mitochondrial matrix protein capable of binding mtDNA. Moreover, we showed that ClpX, the key component of a major mitochondrial protease, interacts with Poldip2 to co-regulate mtDNA elimination in *Drosophila* spermatids. This study sheds light on the mechanisms underlying mtDNA removal during spermatogenesis and underscores the pivotal role of this process in safeguarding maternal inheritance.**

**Keywords** mtDNA Maternal Inheritance; Poldip2; Paternal mtDNA Elimination; Paternal Leakage; ClpXP
**Subject Categories** Development; DNA Replication, Recombination & Repair; Post-translational Modifications & Proteolysis

See also: Z Chen et al

## Introduction

Mitochondria, the major providers of cellular energy, are semi-autonomous organelles that house their own genomes—mitochondrial DNA (mtDNA). Unlike the nuclear genome, which is inherited from both parents, mtDNA is inherited exclusively from the mother in nearly all animal species (Birky, 1995). To date, various mechanisms, including the active degradation of paternal mtDNA during spermatogenesis and/or the subsequent destruction of paternal mitochondria after fertilisation, have been described to ensure maternal mtDNA inheritance in different species (Rantanen et al, 2001; Luo et al, 2013; Nishimura et al, 2006; Chu et al, 2019; Lee et al, 2023; Zhou et al, 2016; Hayashida et al, 2005; DeLuca and O'Farrell, 2012; Yu et al, 2017; Politi et al, 2014; Rojansky et al, 2016; Cummins et al, 1997; Sato and Sato, 2011; Al Rawi et al, 2011; Luo and Sun, 2013).

In vertebrates including fish, mice and humans, there is a huge reduction in mtDNA copy number as sperm mature, highlighting the significant role of mtDNA removal during spermatogenesis in guaranteeing maternal inheritance in these species (Rantanen et al, 2001; Luo et al, 2013; Nishimura et al, 2006; Chu et al, 2019; Lee et al, 2023; Luo and Sun, 2013). Recently, human sperm was found to be devoid of mtDNA, and the disappearance of paternal mtDNA correlates with the relocalisation of TFAM from spermatogonial mitochondria to the spermatozoa nucleus (Lee et al, 2023). This work provides valuable insight into maternal inheritance in humans. However, the regulatory factors governing TFAM relocation remain elusive, raising uncertainties on whether this phenomenon is the cause or the consequence. Moreover, other mechanisms and factors regulating mtDNA elimination in sperm remain largely unexplored. These factors could be mitochondrial-targeted nucleases directly degrading mtDNA (DeLuca and O'Farrell, 2012), or those playing indirect roles through modulating expression or mitochondrial localisation of other proteins.

In *Drosophila melanogaster*, paternal mtDNA elimination predominantly occurs during late spermatid stages. The process is highly efficient and mtDNA is hardly detected in the mature sperm stored in seminal vesicles (DeLuca and O'Farrell, 2012). Like in mammals, *Drosophila* spermatogenesis consists of the differentiation of germline stem cells into spermatogonia cells, mitotic divisions of spermatogonia cells to primary spermatocytes, meiotic divisions of spermatocytes to haploid spermatids, and subsequent spermatid elongation and maturation (Fig. 1A). During early

[1]Gurdon Institute, Tennis Court Road, Cambridge CB2 1QN, UK. [2]Department of Genetics, University of Cambridge, Downing Street, Cambridge CB2 3EH, UK. [3]School of Biosciences, University of Birmingham, Edgbaston B15 2TT, UK. [4]Guangdong Provincial Key Laboratory of Protein Function and Regulation in Agricultural Organisms, South China Agricultural University, Tianhe District, Guangzhou 510642 Guangdong, P. R. China. [5]Present address: Department of Pharmacology, UT Southwestern Medical Center, 6001 Forest Park Road, Dallas, TX 75390, USA. [6]Present address: Institut de Génomique Fonctionnelle de Lyon, ENS de Lyon, CNRS, Univ Lyon 1, 46 Allée d'Italie, Lyon, France. [7]Present address: Cambridge Institute for Medical Research, University of Cambridge, Cambridge CB2 0XY, UK. [8]These authors contributed equally: Ziming Wang, Tirawit Meerod, Nuria Cortes-Silva, Ason C-Y Chiang. ✉E-mail: h.ma.6@bham.ac.uk

spermatid stages, the mitochondrial network undergoes a dramatic transformation with numerous mitochondria fusing to form two large mitochondrial derivatives. As the flagellum elongates, the two derivatives will extend along to fill the entire length of the flagellum (~1.8 mm) (Fuller, 1993). When the flagellum reaches almost its full length, mtDNA elimination occurs, starting at the basal (nuclear) ends and moving to the apical ends. Subsequently, an actin-containing structure called the 'investment cone' moves from the base to the tip and squeezes many cytoplasmic components inside flagella into a waste bag at the end. This investment cone movement separates the spermatids in bundles into individual mature sperm to be stored in seminal vesicles. It also aids in the elimination of residual paternal mtDNA molecules if there are any. EndoG, a mitochondrial endonuclease, was found to be involved in this process. However, the knockdown of *endoG* only delayed the mtDNA clearance, with the remaining paternal mtDNA molecules being removed by the subsequent investment cone movement (DeLuca and O'Farrell, 2012). The knockdown of mtDNA polymerase *polG1* could give rise to mature sperm retaining some mtDNA molecules (Yu et al, 2017). Nevertheless, even in the *polG1* and *endoG* double knockdown males, most mtDNA molecules were still eliminated in late spermatids, and the retained paternal genomes were not detected in progeny. These findings suggest that other factors governing paternal mtDNA elimination are yet to be identified.

In this study, we conducted a large-scale forward genetic screen in *D. melanogaster* and identified 47 mutant lines that retained mtDNA in mature sperm. A gene *poldip2* was revealed by genetic mapping. *poldip2* is mainly expressed in the testis, and its disruption was coupled with significantly elevated levels of mtDNA in mature sperm and a low level of paternal leakage in progeny. Through confocal imaging, proteinase K protection and ChIP assays, we showed that Poldip2 is a mitochondrial matrix protein capable of binding to mtDNA. Finally, we identified ClpX, a key component of the mitochondrial protease ClpXP, as the interacting partner of Poldip2. Knockdown of *clpX* resulted in a similar increase in total mtDNA copy in testis as *poldip2* mutants. Importantly, this aberrant mtDNA copy number was restored to normal levels upon overexpression of Poldip2. Given the temporal alignment of *poldip2* expression with the mtDNA removal period, we propose that the onset of mtDNA elimination is triggered by Poldip2 expression and its binding to mtDNA. This binding event recruits ClpXP to mitochondrial nucleoids, facilitating the digestion of other mtDNA binding proteins and making mtDNA more accessible for degradation by other components. The findings of this study shed light on the mechanisms underlying the clearance of mtDNA during sperm development, emphasising the critical role of this process in safeguarding maternal inheritance.

# Results

## A forward genetic screen identified multiple mutant lines retaining mtDNA in mature sperm

To screen for factors involved in mtDNA elimination in late spermatids, we exposed flies to ethyl methanesulfonate (EMS), a mutagen that introduces random mutations in nuclear DNA. Through genetic crosses, we established ~10,000 lines carrying mutations on the 2nd or 3rd chromosome (Fig. EV1A). Among these EMS lines, 4621 were homozygous viable. We dissected testes/seminal vesicles from these lines and used antibodies against double-stranded DNA (dsDNA) to visualise mtDNA (Fig. 1B). The dsDNA signals co-localise well with mtSSB-GFP (mitochondrial single-stranded binding protein) and PicoGreen (a fluorescent probe that binds dsDNA) signals in spermatocytes and spermatids (Fig. EV1B). In late spermatids where mtDNA replication is decreased, the dsDNA signal remains sharp while mtSSB-GFP and PicoGreen signals become more diffuse. Moreover, the anti-dsDNA antibodies do not stain the highly condensed nuclear DNA of mature sperm, allowing the mtDNA signals in seminal vesicles to be easily discerned for quantification purposes. We categorised the mutants based on the extent of mtDNA retention in their mature sperm stored in seminal vesicles. Through this screen, we identified 47 EMS lines consistently retaining medium to high levels of mtDNA in their mature sperm after multiple rounds of repeats (Fig. 1B,C).

Of the 47 lines, 14 carried the responsible mutations on the 2nd chromosome and 33 on the 3rd chromosome. Sequencing revealed that none of these lines had mutations in *endoG* or *polG1*, indicating that the screen did not reach saturation. This may be partly because we did not examine any homozygous lethal mutants. Complementation crosses among the 3rd chromosome lines identified only four complementary groups, suggesting that the responsible mutations in each EMS line likely affect different genes. Our screen thus indicates that mtDNA elimination during *Drosophila* spermatogenesis is regulated by multiple factors.

## Poldip2 is required for mtDNA elimination in *Drosophila* spermatids

In this study, we focus on one specific line, *EMS-23*, which consistently maintained a medium level of paternal mtDNA in mature sperm (Fig. 2A). Through whole-genome sequencing and subsequent bioinformatics analysis, we identified 27 genes harbouring mutations that could compromise their functions (Appendix Table S1). To map the responsible mutation, we crossed *EMS-23* to the corresponding deficiency lines for 26 genes. None of the transheterozygous progeny showed the mtDNA retention phenotype (Appendix Table S1). For the remaining gene, *poldip2*, deficiency lines were unavailable due to a haploinsufficient gene in the nearby location. Hence, we obtained a hypomorphic mutant ($y^1w^{67c23}$;;$P\{EPgy2\}poldip2^{EY08866}$) from the Bloomington *Drosophila* Stock Center (BDSC) and crossed it with *EMS-23*. *poldip2$^{EY08866}$* carries a P-element insertion in the 5' UTR region of *poldip2*, leading to an approximately 86% reduction in Poldip2 at the protein level (Figs. 2C and EV2A). Both transheterozygotes of *EMS-23/poldip2$^{EY08866}$* and homozygotes of *poldip2$^{EY08866}$* exhibited medium to high levels of mtDNA retention (as shown in Figs. 2A and EV2B), indicating that the missense mutation in *poldip2* is responsible for the mtDNA retention phenotype observed with *EMS-23*.

Poldip2 is a highly conserved protein in metazoans (Hernandes et al, 2017) (Fig. EV2C). It has a YccV-like domain at the N terminus and a DUF525/ApaG domain at the C terminus (Fig. 2B). The YccV-like domain may bind to DNA, whereas the DUF525 domain may be involved in protein-protein interaction and cation efflux (Hernandes et al, 2017). The mutation in our *EMS-23* line

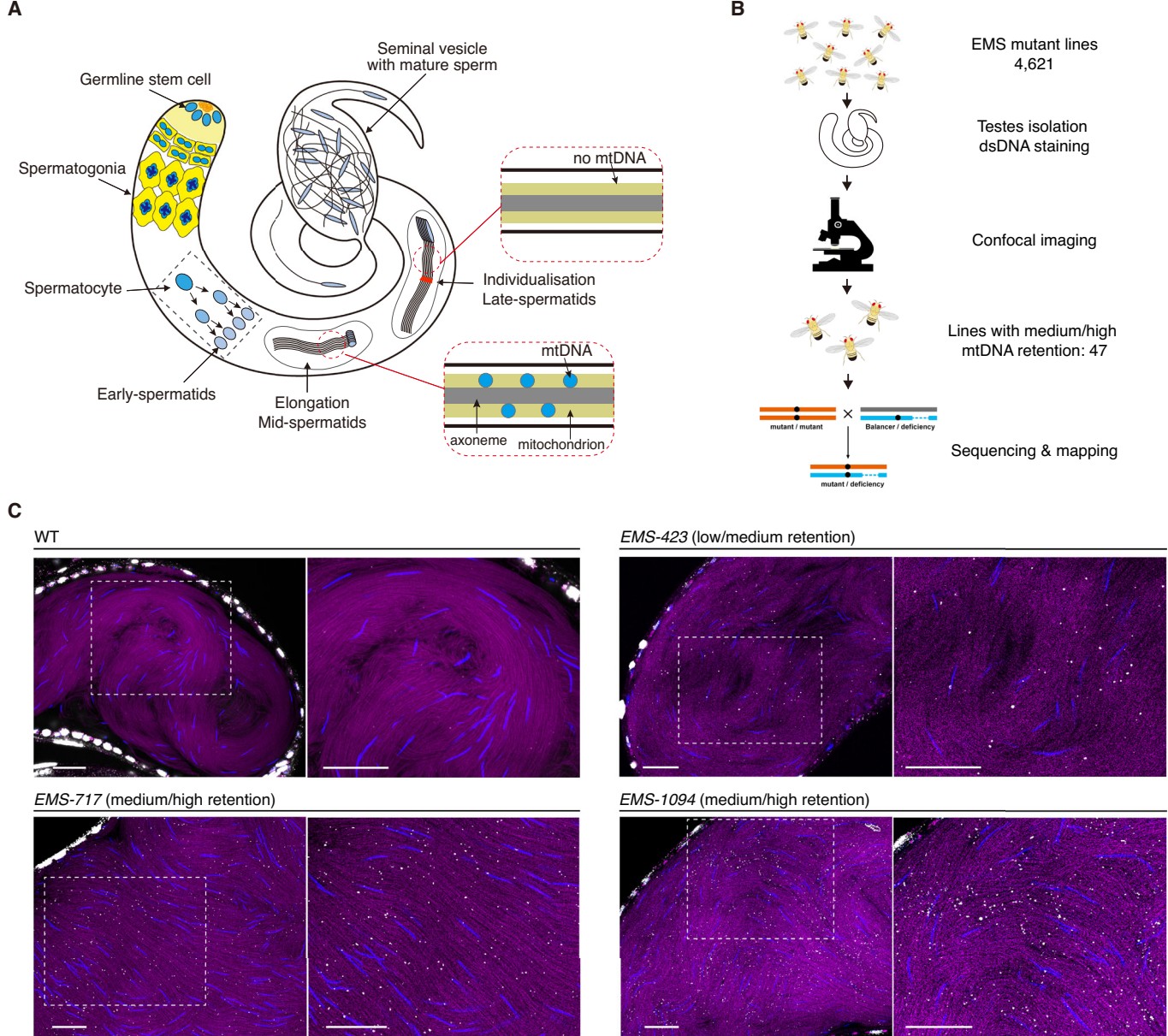

**Figure 1.  A forward genetic screen in *D. melanogaster* identified various mutant lines that retain mtDNA in mature sperm stored in seminal vesicles.**

(A) A schematic representation of spermatogenesis in *D. melanogaster*. The paternal mitochondrial genomes are eliminated in the late spermatid stage before the onset of the individualisation (DeLuca and O'Farrell, 2012). (B) The workflow of the EMS mutagenesis screen to identify lines retaining mtDNA in mature sperm. EMS lines with high mtDNA retention phenotypes were sequenced, followed by bioinformatics analyses and genetic mapping using deficiency and mutant lines to identify the responsible mutations. (C) Representative images of seminal vesicles isolated from wild-type males (a parental line used for EMS mutagenesis) and some EMS lines retaining low to high levels of mtDNA in mature sperm. Mitochondria in magenta (mtSSB-GFP), mtDNA in white (anti-dsDNA antibodies staining), and DAPI in blue. Scale bars: 20 μm. Source data are available online for this figure.

converts a conserved proline to serine in the DUF525 domain (Fig. EV2C). In *Drosophila*, there are two Poldip isoforms, PA and PB, which differ only for the first 58 amino acids preceding the YccV-like domain (Figs. 2B and EV2C). Both isoforms are expressed in the fly testis based on our RNAseq data, with mRNA of PA accounting for 91.39% of the reads. We designed multiple guide RNAs, aiming to isolate CRISPR mutants that affect only one of the isoforms. However, due to the limited sequence disparity between the two isoforms, all mutants we obtained carried deletions

that introduced premature stop codons in both isoforms (Fig. EV2D) and resulted in the complete knockout of *poldip2* (Fig. 2C). These mutants and *poldip2^{KO}/poldip2^{EY08866}* transheter-ozygotes all exhibited medium to high levels of mtDNA in mature sperm (Figs. 2D and EV2E). On average, testes isolated from *poldip2* knockout males harboured ~30% more mtDNA copies than wild type (Fig. 2E). The retention phenotype was significantly reduced upon transgenic expression of either Poldip2-PA or Poldip2-PB isoform (Figs. 2D,E and EV2E). These findings validate

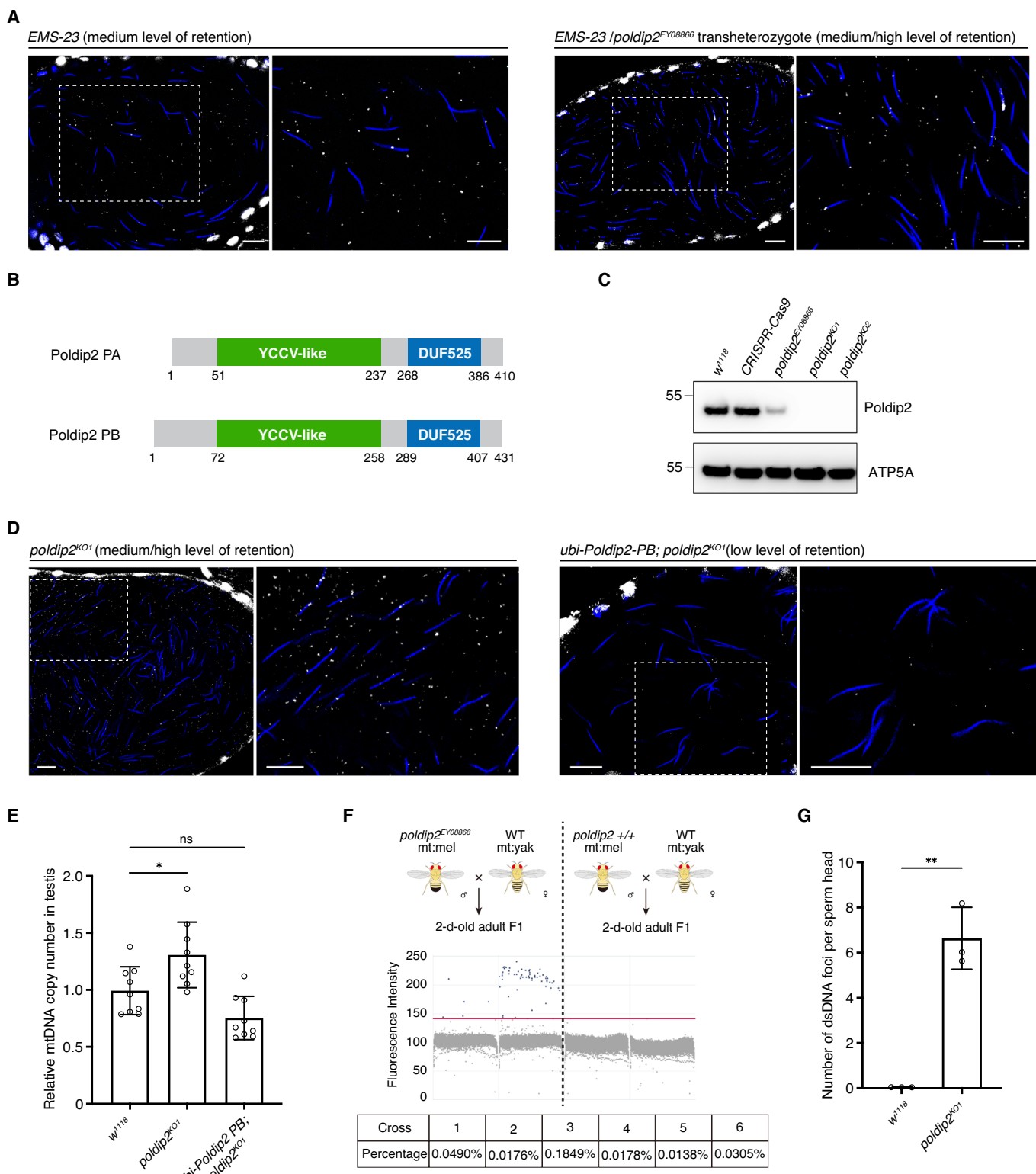

**A** *EMS-23* (medium level of retention)  *EMS-23* /*poldip2^EY08866^* transheterozygote (medium/high level of retention)

**B** Poldip2 PA — YCCV-like / DUF525 — 1 51 237 268 386 410

Poldip2 PB — YCCV-like / DUF525 — 1 72 258 289 407 431

**C** Poldip2 / ATP5A

**D** *poldip2^KO1^* (medium/high level of retention)  *ubi-Poldip2-PB; poldip2^KO1^* (low level of retention)

**E** Relative mtDNA copy number in testis

**F** *poldip2^EY08866^* mt:mel × WT mt:yak → 2-d-old adult F1  *poldip2 +/+* mt:mel × WT mt:yak → 2-d-old adult F1

| Cross | 1 | 2 | 3 | 4 | 5 | 6 |
|-------|---|---|---|---|---|---|
| Percentage | 0.0490% | 0.0176% | 0.1849% | 0.0178% | 0.0138% | 0.0305% |

**G** Number of dsDNA foci per sperm head

the role of Poldip2 in mtDNA removal during *Drosophila* spermatogenesis.

Next, we examined whether the retained paternal mtDNA in *poldip2* mutants could be further transmitted to offspring. To this end, we crossed *poldip2^EY08866^* males carrying wild-type mtDNA to wild-type females homoplasmic for mt:ND2^del1^ and extracted genomic DNA from their adult progeny. The mt:ND2^del1^ variant contains a 9 bp deletion in the mt:ND2 coding region. Previously, we designed primers ending in the deleted region to predominantly amplify wild-type mtDNA and quantify heteroplasmy levels

**Figure 2. Knockout or knockdown of *poldip2* leads to mtDNA retention in mature sperm and a low level of paternal leakage to progeny.**

(A) Representative images of seminal vesicles isolated from *EMS-23* or *EMS-23/poldip2*[EY08866] stained with anti-dsDNA antibodies (white, mtDNA) and DAPI (blue, nuclear DNA). Scale bars: 10 μm. (B) Two Poldip2 isoforms found in *D. melanogaster* and their domains. (C) Western blots to detect Poldip2 levels in various mutants. ATP5A was blotted as the loading control. (D) Representative images of seminal vesicles isolated from *poldip2*[KO1] or *ubi-Poldip2-PB*; *poldip2*[KO1] males stained with anti-dsDNA antibodies (white, mtDNA) and DAPI (blue, nuclear DNA). Scale bars: 10 μm. (E) The relative mtDNA copy number per testis measured by qPCR for *w*[III8], *poldip2*[KO1] or *ubi-Poldip2-PB*; *poldip2*[KO1] flies. The number was normalised to *w*[III8] samples ($n = 9$ biological replicates). Data: mean ± SD, One-Way ANOVA, $P = 0.022$ (*) and 0.098 (ns), respectively. (F) ddPCR plots showing the level of paternal mtDNA in the 2-d-old adult progeny of *poldip2*[EY08866] and wild-type males. *poldip2* mutant or wild-type males carrying *D. melanogaster* mtDNA were crossed to wild-type females homoplasmic with *D. yakuba* mtDNA. The red line marks the threshold, above which droplets were considered positive for *D. melanogaster* mtDNA (i.e. paternal mtDNA). Representative plots from two crosses per group are shown. Paternal leakage levels in *poldip2*[EY08866] progeny across all six crosses are listed. (G) The average number of dsDNA dots per mature sperm of wild-type and *poldip2*[KO1] flies. This number was calculated by quantifying the total number of dsDNA foci in the seminal vesicle and dividing it by the number of sperm heads ($n = 3$ biological replicates). Data: mean ± SD, unpaired Student's t-test, $P = 0.0011$ (**). Source data are available online for this figure.

(Ma et al, 2014). For the six crosses we set up, no evidence of leakage was found based on our qPCR measurement. We then employed digital droplet PCR (ddPCR), a more sensitive method for rare variant detection, to further assess leakage level. Unfortunately, the primer set used to differentiate wild-type mtDNA from mt:ND2[del1] did not offer clear insights, as it could also amplify the mt:ND2[del1] genotype, albeit with significantly lower efficiency. This has little impact on heteroplasmy measurement when the wild-type genome is abundant but poses challenges in distinguishing genuine wild-type amplicons from those generated by mt:ND2[del1] when the wild-type genome is rare, which is expected in this case.

To enhance the sensitivity of our paternal leakage assay, we set up a different cross, where *poldip2*[EY08866] males carrying wild-type mtDNA were mated with *D. melanogaster* females homoplasmic for mtDNA from a closely related species, *Drosophila yakuba* (i.e. *D. melanogaster* (mt:yak)). The *D. yakuba* mtDNA differs from *D. melanogaster* mtDNA by >1000 SNPs/indels. This level of polymorphism enables the design of primers that are highly specific to only one genotype. Additionally, *D. melanogaster* mtDNA has a strong transmission advantage when paired with *D. yakuba* mtDNA (Ma and O'Farrell, 2016). Thus, even if a small amount of paternal *D. melanogaster* mtDNA is leaked to progeny, its transmission advantage will further amplify its abundance to detectable levels. For this cross, we detected paternal leakage in one of the six crosses by qPCR (0.12%). When the more sensitive ddPCR approach was employed, low levels of paternal mtDNA were detected in 2-d-old adult progeny across all six crosses (ranging from 0.014% to 0.185%, with the 0.185% sample being the one that tested positive by qPCR) (Fig. 2F). Such leakage was not observed with progeny of our negative controls, where wild-type males carrying *D. melanogaster* mtDNA were mated with females homoplasmic for *D. yakuba* mtDNA.

By quantifying the total number of dsDNA foci in seminal vesicles and normalising it to the number of sperm heads, we estimated that each mature sperm from *poldip2*[KO1] flies carried an average of 6.6 copies of mtDNA (Fig. 2G). This figure is likely to be an underestimation, as our immunostaining would not capture all dsDNA, and each dsDNA punctum may represent more than one mtDNA molecule. Nevertheless, this suggests that the initial percentage of paternal mtDNA would be around 0.0001%, given that each *Drosophila* egg contains approximately $10^7$ copies of mtDNA (Ma et al, 2014). This is below the detection limits of our ddPCR assay, which lies between 0.005% and 0.0005% (Fig. EV2F). In most of our crosses, paternal *D. melanogaster* mtDNA had

reached roughly 0.01% in 2-d-old adults, representing a 100-fold increase from the embryo to early adult stages. This substantial rise in paternal mtDNA copy number is not unexpected, as we previously demonstrated by cytoplasmic transplantation that *D. melanogaster* mtDNA could outcompete *D. yakuba* mtDNA in two generations (Ma and O'Farrell, 2016). In conclusion, our findings suggest that Poldip2 is essential for mtDNA elimination in *Drosophila* spermatids, and its dysfunction could lead to low levels of paternal leakage.

## Poldip2 is a mitochondrial matrix protein capable of binding mtDNA

Poldip2 was first identified as a binding partner of the DNA polymerase delta p50 subunit and PCNA through a yeast two-hybrid screen using a human cDNA library (Liu et al, 2003). Besides its functions in the nucleus, recent work has revealed an N-terminal mitochondrial localisation sequence of Poldip2 that allows it to localise to the mitochondrial matrix in mammalian cells (Strack et al, 2020; Paredes et al, 2018; Cheng et al, 2005; Arakaki et al, 2006). We confirmed the mitochondria matrix localisation of human Poldip2 in HEK293 cells by cell fractionation and proteinase K protection assays (Fig. EV3A,B).

In *Drosophila*, Poldip2 is predominantly expressed in the testis (Fig. 3A). *poldip2* knockout mutants exhibited >50% reduction in male fertility, whereas female fertility was unaffected (Fig. 3B), highlighting the crucial role of Poldip2 in spermatogenesis. Transgenic overexpression of Poldip2 PA or PB isoforms tagged with mCherry showed clear mitochondrial enrichment in fly testis (Fig. 3C). Close examination revealed that Poldip2 signals were encapsulated by the mitochondrial outer membrane marker Tom20-GFP (Fig. 3D), suggesting that Poldip2 localises in the mitochondrial matrix. Our proteinase K protection assay confirmed this observation as the fly Poldip2 in the crude mitochondrial fraction was resistant to digestion after the outer mitochondrial membrane integrity was disrupted by hypo-osmotic treatment (Fig. 3E).

After confirming that *Drosophila* Poldip2 localises to the mitochondrial matrix, we sought to determine whether Poldip2 co-localises with mtDNA via confocal imaging. However, the high level of Poldip2 expression in the *ubi-Poldip2* transgenic lines and the limited resolution of our confocal microscopy made it challenging to assess the co-localisation. The Poldip2-mCherry signals appeared diffuse overall, although some enrichment at dsDNA foci was observed at the spermatocyte stage (Fig. EV3C).

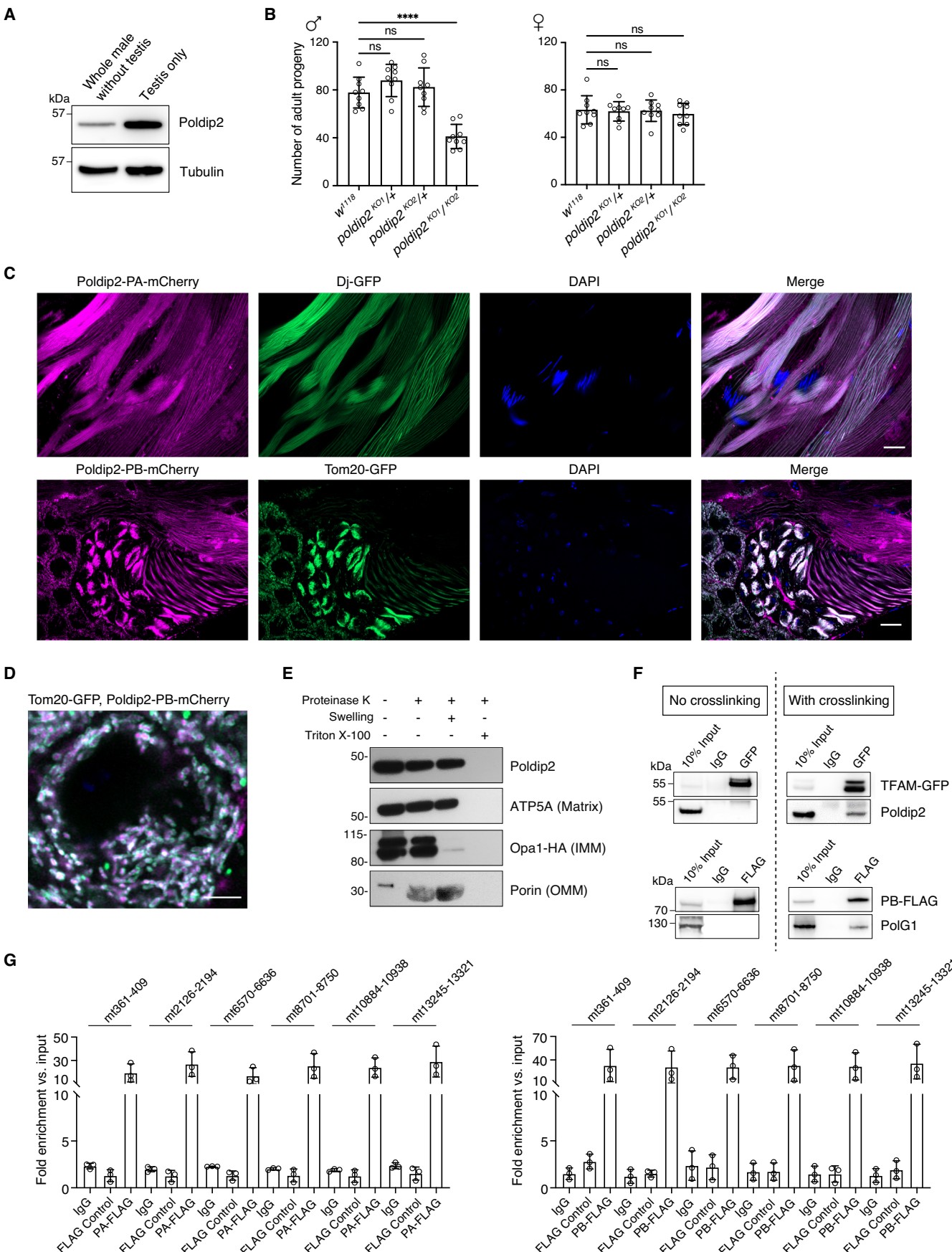

**Figure 3. Poldip2 is a mitochondrial matrix protein and can bind to mtDNA.**

(A) Immunoblots of Poldip2 in the testis and remaining adult male body tissues. Tubulin was probed as the loading control. (B) The number of adult progeny produced by a single male or female parent at 25 °C ($n = 9$ biological replicates). Transheterozygotes of $poldip2^{KO1/KO2}$ were used instead of homozygous $poldip2$ knockout mutants to minimise off-target effects of CRISPR/Cas9-based editing. Data: mean ± SD, One-Way ANOVA, $P = 0.2679$ (ns), 0.8095 (ns), and <0.001 (****), respectively, for males; $P = 0.9826$ (ns), 0.9964 (ns), and 0.7745 (ns), respectively, for females. (C) Representative images showing Poldip2 subcellular localisation in spermatocytes and spermatids. $ubi$-$Poldip2$-$PA/B$-$FLAG$-$mCherry$ transgenic lines were used to visualise Poldip2, whereas $Dj$-$GFP$ and $ubi$-$Tom20$-$GFP$ were used to visualise mitochondria. Scale bars: 20 µm. (D) A confocal image showing Poldip2 (magenta) and Tom20 (green) signals in a spermatocyte of $ubi$-$Poldip2$-$PB$-$FLAG$-$mCherry$; $ubi$-$Tom20$-$GFP$ flies. Scale bar: 5 µm. (E) Immunoblots of the proteinase K protection assay probing $Drosophila$ Poldip2 in different mitochondrial compartments. Porin, Opa1 (endogenously tagged with HA), and ATP5A were probed as the mitochondrial outer membrane (OMM), inner membrane (IMM) and matrix marker, respectively. Embryos were used to obtain sufficient materials. (F) Immunoblots of co-IP with anti-GFP or anti-FLAG antibodies to probe interactions between Poldip2 and TFAM or PolG1 in the testis with or without cross-linking. (G) ChIP-qPCR measuring the mtDNA enrichment levels with Poldip2 immunoprecipitation ($n = 3$ biological replicates). Embryos were used to obtain sufficient materials. Data were normalised to input DNA. IgG control: immunoprecipitating Poldip2-FLAG samples with IgG; FLAG control: immunoprecipitating wild-type samples (i.e. no FLAG expression) with anti-FLAG antibodies to control for unspecific bindings between anti-FLAG antibodies and mtDNA; PA or PB-FLAG: immunoprecipitating Poldip2-FLAG samples with anti-FLAG antibodies. Three independent ChIP assays were performed for both PA and PB isoforms, and the mtDNA enrichment level was measured by qPCR using six pairs of primers binding to different regions of mtDNA. Data: mean ± SD. Source data are available online for this figure.

Hence, we turned to co-immunoprecipitation (co-IP) to examine the interaction between Poldip2 and nucleoid proteins. We detected interactions between Poldip2 and TFAM or mtDNA polymerase PolG1 in testes, but only when cross-linking was applied during sample preparation (Fig. 3F). This suggests that while Poldip2 is unlikely to form a stable interaction with TFAM or PolG1, it may localise near mtDNA nucleoids and weakly associate with these proteins.

Given that the YccV-like domain of Poldip2 has DNA-binding potential, we next tested whether Poldip2 could bind to mtDNA. To address this question, we conducted Chromatin Immunoprecipitation (ChIP) assays on transgenic flies expressing Poldip2-PA/B-FLAG-mCherry followed by qPCR. Our analysis revealed ~25-fold and 35-fold enrichment of mtDNA with PA and PB immunoprecipitation, respectively (Fig. 3G). Notably, the enrichment level was consistent when primers binding to different regions of mtDNA were used for qPCR, suggesting that Poldip2 binds to mtDNA uniformly. In parallel, we conducted ChIP assays on TFAM-GFP flies as a positive control because TFAM is known to bind broadly across the mitochondrial genome with high affinity and quantities (Wang et al, 2013). The enrichment of mtDNA with TFAM immunoprecipitation was ~180 folds based on three replicas (Fig. EV3D). The observed differences in mtDNA enrichment between TFAM and Poldip2 may result from variations in their mtDNA binding affinities and quantities, or differences in the affinity and sensitivity of the GFP and FLAG antibodies used in our ChIP assay. In summary, our data support the conclusion that both Poldip2 PA and PB isoforms are mitochondrial matrix proteins capable of binding mtDNA.

## ClpX interacts with Poldip2 and likely co-regulates mtDNA elimination

Although capable of binding to mtDNA, Poldip2 does not appear to regulate mtDNA transcription, as our RNAseq data showed that the knockdown of $poldip2$ had little impact on mRNA levels of mtDNA genes in the testis (Fig. EV4A). Knockout of $poldip2$ also did not cause an obvious change in mtDNA nucleoid sizes at various stages of spermatogenesis (Fig. EV4B). To investigate how Poldip2 regulates mtDNA elimination in $Drosophila$ spermatids, we probed its binding partners through pull-down assays with $ubi$-$Poldip2$-$PB$-$FLAG$-$mCherry$ flies followed by mass spectrometry

analysis. Through this, we identified ClpX as one of the interactors (Fig. 4A). The Poldip2-PA isoform also interacted with ClpX, as shown by our co-IP assay (Fig. EV4C).

ClpX forms a complex with ClpP called ClpXP, which is one of the major proteases found in the mitochondrial matrix in eukaryotes. It has been demonstrated to interact with Poldip2 in two previous studies using human cells (Paredes et al, 2018; Strack et al, 2020). In particular, Strack et al showed that Poldip2 docks to the N-terminal C4-type zinc-finger domain of ClpX, and this binding stabilises ClpX by protecting it from Lon-mediated degradation. They further revealed that Poldip2 is neither a substrate of ClpXP nor does it cause dissociation of the ClpXP complex. Rather, it acts as an adaptor for ClpXP, which might facilitate the binding and delivery of a substrate to the ClpXP complex (Strack et al, 2020). Indeed, we found that the ClpX level was reduced in all our $poldip2$ mutants by >35%, whereas the knockdown of $clpX$ had minimal impact on Poldip2 levels (Fig. 4B,C).

In human cells, the knockdown of ClpXP has been associated with abnormalities in mtDNA nucleoid packaging and numbers (Key et al, 2021; Torres-Odio et al, 2022). Hence, we speculated that ClpXP co-regulates mtDNA removal alongside Poldip2 in $Drosophila$ spermatids. To test this hypothesis, we obtained a $clpX$ hypomorphic mutant from BDSC ($y^1$ $w^*$;;$P\{Mae$-$UAS.6.11\}$ $clpX^{LA00797}$) (Fig. EV4D), which carries a P-element insertion in the 5' UTR region that reduces the ClpX protein level by ~85% (Fig. 4C). Similar to $poldip2$ mutants, $clpX^{LA00797}$ exhibited medium to high levels of mtDNA retention in mature sperm, and a ~30% increase in total mtDNA copy numbers in the testis (Fig. 4D,E). Knocking down $clpX$ by expressing RNAi from early spermatogenesis ($nos$+$Bam$-$GAL4$) resulted in a comparable level of mtDNA retention in seminal vesicles (Fig. EV4E). Notably, the retention phenotype and the abnormal mtDNA copy number observed with $clpX^{LA00797}$ could be rescued by Poldip2-PB overexpression (Figs. 4E and EV4F), potentially through stabilising more ClpXP complex in the $clpX$ knockdown mutant (Fig. 4F). These data suggest that Poldip2 and ClpXP work together to regulate mtDNA elimination in spermatids.

Despite numerous attempts, we failed to generate an endogenously tagged line to visualise the timing of Poldip2 expression during spermatogenesis. However, Chen et al (2025) successfully generated such a line and revealed that endogenous Poldip2 starts

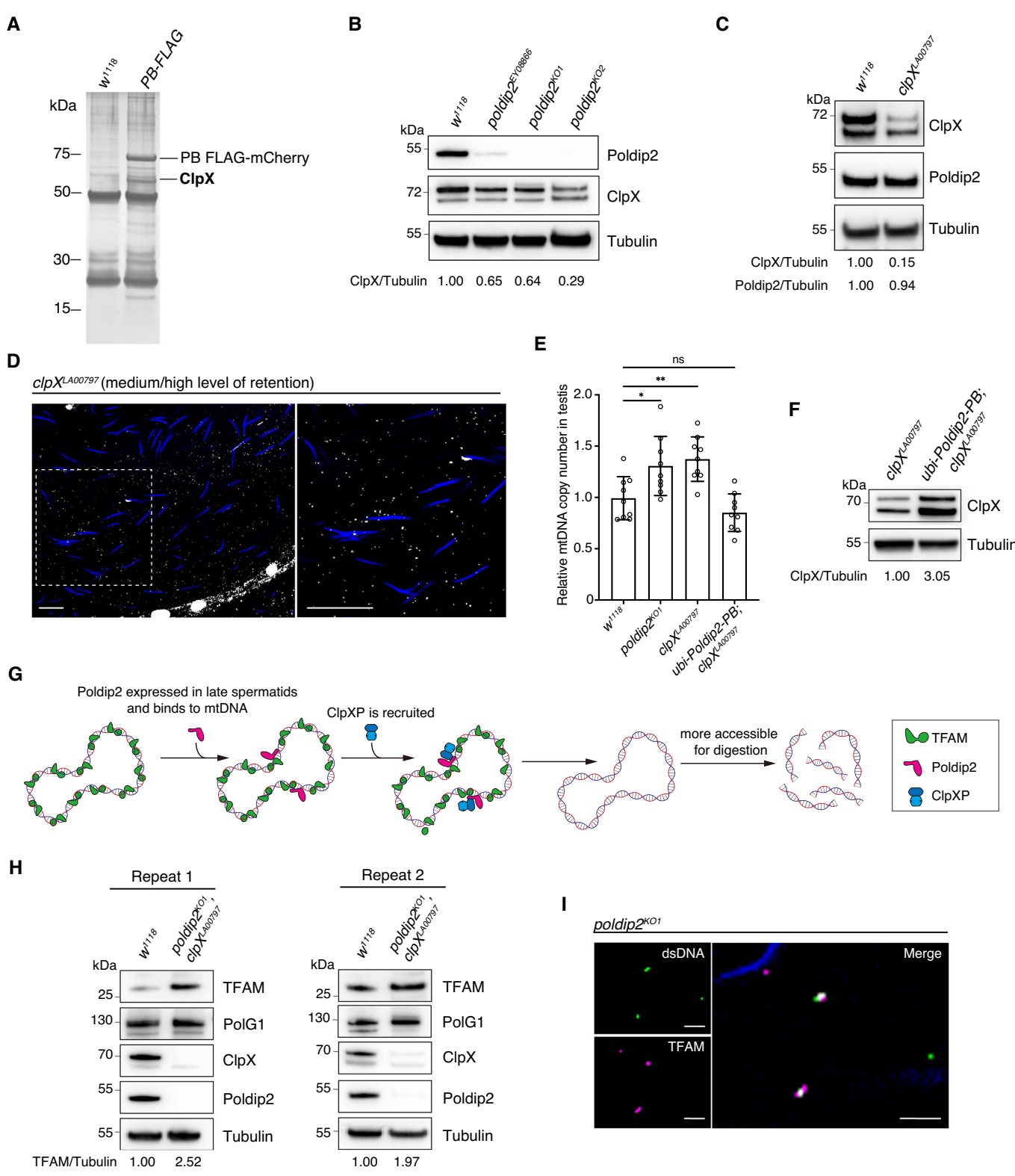

**G** Poldip2 expressed in late spermatids and binds to mtDNA. ClpXP is recruited. more accessible for digestion. TFAM, Poldip2, ClpXP

to express around the mid-spermatid stage. Hence, we speculate that Poldip2 starts to express and bind to mtDNA just before the onset of the mtDNA elimination. This binding event recruits ClpXP to the mtDNA nucleoid region, facilitating the digestion of mtDNA-binding proteins such as TFAM. Consequently, mtDNA becomes more accessible for digestion by other factors such as EndoG (Fig. 4G). In support of this model, our immunoblots showed the TFAM level was doubled in the *poldip2* and *clpX* double

◄ **Figure 4.  ClpX and Poldip2 co-regulate mtDNA elimination in *Drosophila* spermatids.**

(A) Silver staining of a pulldown gel identified ClpX as one of the interactors of Poldip2 in the testis. Three independent pulldown experiments were conducted using anti-FLAG antibodies and *ubi-Poldip2-PB-FLAG-mCherry* flies (see Source data for the other two biological repeats). (B) Immunoblots probing ClpX levels in testes of *poldip2* mutants. Tubulin was blotted as a loading control for calculating the level of ClpX reduction. (C) Immunoblots showing the ClpX and Poldip2 levels in *clpX*$^{LA00797}$ testes. Tubulin was blotted as a loading control for calculating the level of reduction for ClpX and Poldip2. (D) A representative image of *clpX*$^{LA00797}$ seminal vesicles stained with anti-dsDNA antibodies (white) and DAPI (blue). Scale bars: 10 µm. (E) The relative mtDNA copy number per testis for $w^{TII8}$, *poldip2*$^{KO1}$, *clpX*$^{LA00797}$ and *ubi-Poldip2-PB*; *clpX*$^{LA00797}$ males measured by qPCR. The number was normalised to $w^{TII8}$ samples ($n = 9$ biological replicates). Data: mean ± SD, One-Way ANOVA, $P = 0.0144$ (*), 0.0045 (**) and 0.3886 (ns), respectively. The data used to plot Fig. 2E are employed here for the $w^{TII8}$ and *poldip2*$^{KO1}$ samples. (F) Immunoblots of ClpX in *clpX*$^{LA00797}$ and *ubi-Poldip2-PB*; *clpX*$^{LA00797}$ flies. Tubulin was blotted as a loading control. (G) A model depicting a potential mechanism on how Poldip2 and ClpXP govern mtDNA removal in *Drosophila* spermatids. (H) Immunoblots of TFAM and PolG1 in *poldip2*$^{KO1}$, *clpX*$^{LA00797}$ testes (two biological repeats). Tubulin was blotted as a loading control. (I) A representative image of *poldip2*$^{KO1}$ mature sperm stored in seminal vesicles stained with antibodies against dsDNA (green) and *Drosophila* TFAM (magenta). Nuclear DNA was labelled by DAPI (blue). Scale bar: 2.5 µm. Source data are available online for this figure.

mutant, whereas the PolG1 level remained comparable to the wild type (Fig. 4H). Furthermore, many dsDNA dots in the seminal vesicle of *poldip2* mutants co-localised with TFAM signals, indicating that retained mtDNA molecules were still associated with TFAM (Fig. 4I). This suggests that Poldip2 could be required to remove TFAM from mtDNA nucleoids in spermatids. Together, these observations underscore the critical role of Poldip2 and ClpXP in regulating mtDNA dynamics during *Drosophila* spermatogenesis.

## Discussion

This work identified Poldip2 through a forward genetic screen to regulate mtDNA elimination in *Drosophila* spermatids. Compromising Poldip2 resulted in mtDNA retention in mature sperm and low levels of paternal leakage to progeny. We further showed that Poldip2 is a mitochondrial matrix protein capable of binding to mtDNA. Its interactor, ClpXP protease, is likely to co-regulate mtDNA elimination alongside Poldip2 to ensure the maternal inheritance of mtDNA in *D. melanogaster*.

The expression of Poldip2 coincides with the timing of mtDNA removal (as indicated by Chen et al, 2025), suggesting that the onset of mtDNA elimination may be triggered by the appearance of Poldip2 in spermatids. This expression pattern might be controlled at the transcriptional or translational level, with *poldip2* mRNA or protein being made only at this developmental stage. Another possibility is that Poldip2 is continuously degraded at earlier stages via post-translational regulation. Our mass spectrometry data revealed several potential sites for post-translational modifications on the Poldip2-PB isoform (MS1 spectra: phosphorylation at Y73, T75, S110, T112, T113, T204, and T307, and carbamidomethylation at C197, C304 and C398; with T112/113 and T204 phosphorylation sites confirmed by MS2 spectra) (Fig. EV5A). Modifications at these sites may affect Poldip2 stability, its interactions with other proteins, and its functional roles across various tissues and developmental stages.

In humans, there are also two Poldip2 isoforms (37 kDa and 42 kDa). Our work in flies showed that although PA seems to have a higher expression based on the RNAseq counts, both isoforms can localise to the mitochondrial matrix, bind to mtDNA and interact with ClpX. Moreover, overexpression of either isoform is sufficient to rescue the mtDNA retention phenotype observed in *poldip2* mutants where both isoforms were knocked out (Figs. 2D,E and EV2E). These findings suggest the two isoforms perform

redundant functions inside mitochondria. However, since we were unable to generate mutants affecting only one isoform, the possibility that one isoform may have a more prominent role in mtDNA elimination cannot be ruled out. Additionally, the two isoforms may play different roles in the nucleus.

Our study also established a connection between Poldip2 and ClpXP in regulating mtDNA elimination in spermatids. ClpXP primarily functions in degrading misfolded or damaged proteins inside mitochondria. It is a barrel-shaped hetero-oligomeric complex, in which ClpP forms a two-stack heptameric ring-shaped structure to which two hexameric ClpX rings bind to each side (Gatsogiannis et al, 2019). Unlike Poldip2, ClpX is expressed from early spermatogenesis, continuing until the very late spermatid stages when Dj-GFP expression becomes prominent, which is a marker for fully elongated spermatid bundles or individualised sperm (Santel et al, 1997) (Fig. EV5B). This divergence in expression timing compared to Poldip2 suggests that ClpX has broader roles in mitochondrial function beyond mtDNA elimination.

The range of substrates targeted by ClpXP remains largely unexplored. In *Drosophila* S2 cells, ClpXP was found to regulate the abundance of mitochondrial mRNA by selectively degrading mLRPPRC1 (Matsushima et al, 2017). In human and murine cells, mutating ClpP resulted in the accumulation of numerous proteins including ASS1, ACADVL, STOM, PRDX3, UQCRC2 and ACADSB (Key et al, 2021). In this study, we showed that Poldip2 levels in the fly testis were not affected in the *clpX* mutant, whereas ClpX levels were positively correlated to Poldip2 levels (Fig. 4B,F). This observation supports Strack et al's finding that Poldip2 is not a substrate of ClpXP, and binding to Poldip2 protects ClpX from being digested by another mitochondrial protease Lon (Strack et al, 2020). We further showed that the TFAM level was increased in the testis of the *poldip2* and *clpX* double mutant (Fig. 4H). In addition, our co-IP assays detected interactions between ClpX and TFAM when cross-linking reagents were used (Fig. EV5C). These findings raise the possibility that TFAM might be a substrate of ClpXP. Altogether, our findings reveal a complex regulatory interaction between Poldip2, ClpX, and TFAM during *Drosophila* spermatogenesis. Further investigation with an increased spatial and temporal resolution is needed to better understand this relationship.

Besides mtDNA elimination in spermatogenesis, the paternal mitochondria are degraded in embryos after fertilisation via autophagy and other pathways in many animal species (Nishimura et al, 2006; Rojansky et al, 2016; Luo et al, 2013; Cummins et al,

1997; Luo and Sun, 2013). This process is believed to further eliminate paternal mitochondrial genomes, particularly in cases where the sperm that has entered the egg still harbours a small amount of mtDNA (Luo et al, 2013; Nishimura et al, 2006; Chu et al, 2019; Luo and Sun, 2013). In *C. elegans*, paternal mtDNA removal relies mainly on such post-fertilisation mechanisms (Sato and Sato, 2011; Al Rawi et al, 2011; Zhou et al, 2016), albeit a recent study showed that sperm also export healthy mitochondria to the extracellular environment via exocytosis to reduce mitochondria numbers prior to fertilisation (Liu et al, 2023). In *D. melanogaster*, paternal mitochondria are ubiquitinated through a lysine 63-linked polyubiquitin side chain and degraded soon after fertilisation (Politi et al, 2014). Mutations in genes including *atg7* and *rab7* attenuate the degradation, indicating the involvement of autophagic and endocytic pathways in this process (Politi et al, 2014). Nevertheless, paternal leakage has been reported in various populations, particularly with inter-species crosses (Kondo et al, 1990; Kvist et al, 2003; Kraytsberg, 2004; Khan et al, 2019; Zhao et al, 2004; Gyllensten et al, 1991; Polovina et al, 2020). This heightened incidence of leakage in inter-species crosses may be partly attributed to the high sequence polymorphisms between the two parental mtDNA genotypes, facilitating the detection of their co-existence by PCR-based method. The detection of paternal mtDNA could be further enhanced if the paternal mtDNA has a selective advantage that allows it to increase abundance. For example, Cao et al reported that paternal mtDNA was more likely to be detected when females carrying detrimental mtDNA mutations (Cao et al, 2024).

In our setup to assess paternal leakage, we integrated both high sequence polymorphisms of parental genomes and transmission advantage of the paternal genome to detect low levels of paternal leakage in adult progeny when *poldip2* mutants were used as male parents (Fig. 2F). This finding suggests that some carried-over paternal mtDNA copies could escape the post-fertilisation elimination and persist to the adult stage, especially when they have transmission advantages. To follow up on this, we further examined paternal leakage using female parents overexpressing a dominant negative *rab7* mutation - *rab7^{T22N}*, which has been shown to delay the post-fertilisation of paternal mitochondrial degradation in *Drosophila* (Politi et al, 2014). The overall paternal mtDNA levels in the progeny appeared slightly higher than those in the progeny of wild-type female parents (Fig. EV5D). However, large variations among individual crosses within each group made it difficult to accurately assess the capacity of post-fertilisation removal mechanisms in purging carried over mtDNA molecules (Fig. EV5D). Future experiments with large sample sizes will offer more insights.

It is important to note that even in *poldip2* knockout mutants, there was a significant reduction in mtDNA foci number in late spermatids compared to those in early elongation stages. This decline is probably facilitated by the investment cone, which is known to remove a substantial portion of residual mtDNA molecules (DeLuca and O'Farrell, 2012). Furthermore, neither knockout nor overexpression of *poldip2* in somatic tissues resulted in changes to total mtDNA copy numbers, suggesting that Poldip2's effect on mtDNA levels might require certain testis-specific components (Fig. EV5E). These observations, alongside our EMS screen that isolated over 40 mutant lines that retain mtDNA in their mature sperm, clearly indicate that additional factors

regulating mtDNA elimination in *Drosophila* sperm remain to be identified.

In conclusion, our work sheds light on the mechanisms underlying mtDNA removal during *Drosophila* spermatogenesis. The detailed processes driving mtDNA elimination appear to differ between species. For instance, in *Drosophila*, TFAM does not translocate from the mitochondria to the nucleus in mature sperm (Fig. EV5F), as observed in humans (Lee et al, 2023). Yet, similar components may be involved in mtDNA elimination at certain stages of spermatogenesis. Considering the high conservation of Poldip2 and ClpX, it is worth investigating whether they perform similar functions to guarantee the maternal inheritance of mtDNA in other species. This will deepen our knowledge of mtDNA biology and provide broader insights into reproductive and evolutionary processes across species.

# Methods

**Reagents and tools table**

| Reagent/Resource | Reference or Source | Identifier or Catalog Number |
|---|---|---|
| **Experimental models** | | |
| *w^{1118}* | Bloomington stock center | BDSC3605 |
| *nos-Cas9* | Bloomington stock center | BDSC54591 |
| EMS parental line 2L (*FRT40A; sqh-mitoYFP*) | This lab | N/A |
| EMS parental line 2R (*FRTG13; sqh-mitoYFP*) | This lab | N/A |
| EMS parental line 3L (*ubi-mtSSB-GFP; FRT2A*) | This lab | N/A |
| EMS parental line 3R (*ubi-mtSSB-RFP; FRT82B*) | This lab | N/A |
| *nos-GAL4, Bam-GAL4* | This lab | N/A |
| *D. melanogaster* (mt:yak) | This lab | N/A |
| *P{EPgy2}poldip2^{EY08866}* | Bloomington stock center | BDSC17500 |
| *P{Mae-UAS.6.11}clpX^{LA00797}* | Bloomington stock center | BDSC22252 |
| *clpX* RNAi (*P{TRiP.HMJ21681}*) | Bloomington stock center | BDSC52989 |
| *Dj-GFP* | Bloomington stock center | BDSC5417 |
| *ubi-mtSSB-GFP* | This lab | N/A |
| *TFAM-GFP* (BAC line) | (Zhang et al, 2016) | N/A |
| *ubi-Tom20-GFP* | This study | N/A |
| *poldip2^{EY08866}* | Bloomington stock center | BDSC17500 |
| *poldip2^{KO1}* (*poldip2^{Δ340-343}*) | This study | N/A |
| *poldip2^{KO2}* (*poldip2^{Δ275, Δ278}*) | This study | N/A |
| *poldip2^{KO3}* (*poldip2^{Δ267-345}*) | This study | N/A |
| *ubi-Poldip2-PA-FLAG-mCherry* | This study | N/A |

| Reagent/Resource | Reference or Source | Identifier or Catalog Number |
|---|---|---|
| *ubi-Poldip2-PB-FLAG-mCherry* | This study | N/A |
| *UASp-YFP-rab7^{T22N}* | Bloomington stock center | BDSC9778 |
| *mata4-GAL4-VP16* | Bloomington stock center | BDSC7063 |
| **Recombinant DNA** | | |
| *pUbi-Poldip2-PA-FLAG-mCherry* | This study | N/A |
| *pUbi-Poldip2-PB-FLAG-mCherry* | This study | N/A |
| *pCFD5-poldip2-gRNA* | This study | N/A |
| **Antibodies** | | |
| Mouse anti-dsDNA | Abcam | Ab27156 |
| Mouse anti-Tubulin | DSHB | 12G10 |
| Mouse anti-ATP5A | Abcam | ab14748 |
| Mouse anti-TFAM | Abcam | ab119684 |
| Rabbit anti-dmTFAM | This study | N/A |
| Mouse anti-FLAG | Merck | F3165 |
| Chicken anti-GFP | Abcam | ab13970 |
| Rabbit anti-GFP | Antibodies | A290 |
| Rabbit anti-ClpX | ThermoFisher Scientific | PA5-79052 |
| Rabbit anti-Poldip2 | Abcam | ab181841 |
| Rabbit anti-Poldip2 | Proteintech | 15080-1-AP |
| Rabbit anti-PolG1 | Abcam | ab128899 |
| Rabbit anti-Tom20 | Abcam | ab186735 |
| Rabbit anti-Tim50 | Proteintech | 22229-1-AP |
| Rabbit anti-dmPolG1 | (Yu et al, 2017) | N/A |
| Rabbit anti-HA | Cell Signaling Technology | 3724S |
| Goat anti-Rabbit 488 | ThermoFisher Scientific | A11008 |
| Goat anti-Rabbit 647 | ThermoFisher Scientific | A21244 |
| Goat anti-Mouse 488 | ThermoFisher Scientific | A1001 |
| Goat anti-Mouse 647 | ThermoFisher Scientific | A221236 |
| Goat anti-Chicken IgY (HRP) | ThermoFisher Scientific | A16054 |
| Goat anti-Mouse IgG | ThermoFisher Scientific | 62-6520 |
| Goat anti-Rabbit IgG-HRP | Novus Biologicals | HAF008 |
| **Oligonucleotides and other sequence-based reagents** | | |
| PCR primers | This study | Appendix Table S2 |
| **Chemicals, Enzymes and other reagents** | | |
| Mouse IgG | Merck | I5381 |
| Rabbit IgG | Cell Signaling Technology | 2729 |
| Proteinase K | New England Biolabs | P8107S |

| Reagent/Resource | Reference or Source | Identifier or Catalog Number |
|---|---|---|
| VECTASHIELD | Vector Laboratories | H-1200 |
| InFusion Cloning Kit | Takara Bio | 638938 |
| Illumina DNA Prep kit | Illumina | 20018705 |
| Illumina index kit | Illumina | 20027213 |
| LDS sample buffer | Merck | MPSB |
| Bolt sample reducing agent | ThermoFisher Scientific | B0009 |
| mPAGE 4–12%, Bis-Tris gel | Merck | MP41G12 |
| MES SDS running buffer | ThermoFisher Scientific | B0002 |
| Immobilon-P PVDF membranes | Merck | IPVH00010 |
| Clarity Western ECL substrate | Bio-Rad Laboratories | 1705061 |
| cOmplete protease inhibitor cocktails | Roche | 11697498001 |
| Dynabeads protein G beads | Invitrogen | 10004D |
| SensiFast SYBR PCR Master Mix | Bioline | 98020 |
| SimpleChIP Plus Kit protocol | Cell Signaling Technology | 9005 |
| Protease inhibitor cocktail | Merck | 11836170001 |
| Qubit protein assay kit | ThermoFisher Scientific | Q33211 |
| QIAcuity EG PCR Kit | QIAGEN | 250113 |
| TRIzol | ThermoFisher Scientific | 15596026 |
| RNase-free DNase I | New England Biolabs | M0303 |
| Next mRNA Magnetic Isolation Module | New England Biolabs | E7490L |
| Next Ultra Directional RNA Library Prep Kit | New England Biolabs | E7760 |
| AMPure XP Beads | Beckman Coulter | A63881 |
| Pierce™ Silver Stain Kit | ThermoFisher Scientific | 24612 |
| **Software** | | |
| GitHub Script for EMS mapping | This study | (https://github.com/adamjamesreid/dmvar) |
| Prism 9 | Graphic Pad | N/A |

## Fly stocks and husbandry

All fly stocks were raised on standard propanoic media at 25 °C unless otherwise stated. Lines used in this study are listed in the Reagents and Tools Table. To generate *poldip2* mutants, plasmids expressing gRNAs were injected into *nos*-Cas9 flies. Individual lineages were established from progeny by genetic crosses, and the nature of mutations was determined by Sanger sequencing. To generate the *ubi-Poldip2-FLAG-mCherry* transgenic lines, the cDNA sequences of two Poldip2 isoforms were fused with 3xFLAG

and mCherry, and cloned into a pUbi vector via the InFusion Cloning Kit (Takara Bio, 638938). Plasmids were injected into *vas-int; attP40* flies to establish the stocks.

## EMS mutagenesis

The following genotypes were used for the EMS mutagenesis screen: [*FRT40A*; *sqh-mitoYFP*], [*FRTG13*; *sqh-mitoYFP*], [*ubi-mtSSB-GFP*; *FRT2A*], and [*ubi-mtSSB-RFP*; *FRT82B*]. First, 2-d-old males were starved at room temperature for 8 h and then fed with 1% sucrose solution containing 25 mM of EMS for 16 h. These males were then group-mated to virgin females with a balancer chromosome ([*Kr^{If}/CyO*; *sqh-mitoYFP*], [*ubi-mtSSB-GFP*; *MKRS/TM6B*], or [*ubi-mtSSB-RFP*; *MKRS/TM6B*]) for 3 days. The males were then removed, leaving mated females to continue egg-laying. The F1 male progeny were then single-mated to virgin females with the corresponding balancer and mitochondrial marker mentioned above to establish individual lines. Approximately, 2500 lines were established for each chromosome arm, generating 10,000 EMS lines carrying mutations on either the 2nd or 3rd chromosome.

## Whole-genome sequencing and bioinformatics analysis to identify mutations in EMS lines

To extract genomic DNA for sequencing, 10 adult males were homogenised in 500 μl of solution A buffer (0.1 M Tris-HCl pH 7.5, 0.01 M EDTA pH 8.0, 1% SDS) and incubated for 20 min at 70 °C. 70 μl of 8 M potassium acetate was then added to the samples and incubated for 20 min on ice. After the incubation, the samples were centrifuged for 15 min at maximum speed. The supernatant was mixed with 250 μl of isopropanol and centrifuged for 4 min at 13,000 × *g*. The pellet was washed with 70% ethanol and diluted in 100 μl ddH$_2$O. The DNA library was prepared using the Illumina DNA Prep kit (Illumina, 20018705) and index kit (Illumina, 20027213) following the manufacturer's instructions. Paired-end reads from Illumina NovaSeq6000 were aligned against the *D. melanogaster* genome (BDGP Release 6 + ISO1MT/dm6).

An in-house bioinformatics pipeline was developed to identify mutations in each EMS line. The code and instructions for running the analysis can be found on the dmvar GitHub page (https://github.com/adamjamesreid/dmvar). In brief, the sequencing batches were processed using nf-core/sarek v2.7.1 (PMID: 32269765) and mapped to the iGenomes dm6 reference (--genome dm6) with the options --tools haplotypecaller --generate_gvcf, which generated GATK GVCFs for each sample. Subsequently, batches of GVCFs, including relevant parental samples from different sequencing batches, were combined and filtered using a custom NextFlow pipeline called dmvar.nf (https://github.com/adamjamesreid/dmvar). This pipeline utilised gatk CombineGVCFs and gatk GenotypeGVCFs (PMID: 21478889) to produce a combined VCF file. SNP and indel calls were then separated using gatk SelectVariants. SNPs were further filtered using gatk VariantFiltration with the following options: 'QD < 2.0' --filter-name 'QD2', 'QUAL < 30.0' --filter-name 'QUAL30', 'SOR > 3.0' --filter-name 'SOR3', 'FS > 60.0' --filter-name 'FS60', 'MQ < 40.0' --filter-name 'MQ40', 'MQRankSum < -12.5' --filter-name 'MQRankSum-12.5', and 'ReadPosRankSum < -8.0' --filter-name 'ReadPosRankSum-8'. Indels were filtered using the following options: 'QD < 2.0' --filter-name 'QD2', 'QUAL < 30.0' --filter-

name 'QUAL30', 'FS > 200.0' --filter-name 'FS200', and 'Read-PosRankSum < -20.0' --filter-name 'ReadPosRankSum-20'. Variant annotation was performed using snpEff ann (v5.0; PMID: 22728672) with reference BDGP6.28.99. The separate VCF files were then merged using bcftools concat with overlapping variants (-a; PMID: 33590861).

A custom script called dmvar_results.py (available on GitHub) was utilised to generate an Excel file for each parental control, with each mutant having its own sheet detailing the variants of interest. For each cohort (parent and progeny), the variants were filtered to include only those that were heterozygous or homozygous, depending on the sample. A variant site had to be homozygous in the parent, and both homozygous and heterozygous alternative calls were considered in the offspring. Homozygous calls in the offspring were allowed to have relatively low frequencies due to potential noise in sample preparation, thus a minimum alternative allele frequency of 0.6 was used. The results were further annotated using gene locations and descriptions from FlyBase (PMID: 35266522; automated_gene_summaries.tsv, downloaded December 2021; gene_map_table_fb_2021_06.tsv.gz).

## Immunofluorescence and confocal microscopy imaging

Testes dissected from homozygous viable EMS lines were fixed in phosphate-buffered saline (PBS) with 4% paraformaldehyde (pH 7.4) for 30 min, washed three times in PBS, and permeabilised with PBST (PBS + 0.1–1% Triton X-100) for 1 h at room temperature. The samples were then blocked in PBST supplemented with 3% bovine serum albumin for 1 h. If subsequent immunostaining was required, testes were incubated with the following antibodies: dsDNA (mouse; Abcam, ab27156, 1:200), *Drosophila* TFAM (rabbit, custom-made, 1:1000), or ClpX (rabbit; ThermoFisher Scientific, PA5-79052, 1:200) overnight at 4 °C, followed by another overnight incubation with secondary antibodies goat anti-mouse IgG or anti-rabbit Alexa Fluor 488 and 647. Samples were then mounted in VECTASHIELD with DAPI (Vector Laboratories, H-1200) and imaged on a Leica SP8 or Zeiss LSM900 confocal microscope. Antibodies against *Drosophila* TFAM were generated by Hangzhou HuaAn Biotechnology Co., Ltd using the following antigen sequence: PYFRFMREQRPKLKAANP-QITTVEVVRQLSKNWSDADAQLKERLQAEFKRDQQIYVEERTK YDATLTEEQRAEIKQLKQDLVDAKERRQLRKRVKELGRPKKPA-SAFLRFI.

## Genetic mapping

To map the mutation(s) responsible for the mtDNA retention phenotype in individual EMS lines, three 2-d-old males from the corresponding deficiency/mutant line were mated with five virgin females of the EMS mutants at 25 °C. Testes from five 1–3-d-old male progeny were dissected for immunostaining and confocal imaging as described above. Crosses generating progeny with mtDNA retention phenotypes were repeated at least three times.

## Fertility test

Males of *poldip2* knockout lines were crossed to *w^{1118}* females with balancer chromosomes to establish mutant lines with an isogenic nuclear background (except for the 3rd chromosome). To test the male fertility, individual 2-d-old males were mated with three *w^{1118}*

virgin females at 25 °C, and the total number of progeny was counted for each cross. For the female fertility assay, two 2-d-old females were mated with three young $w^{1118}$ males at 25 °C, and the total number of progeny was counted for each cross. For each genotype, 9 replicates were performed.

## Western blot analysis

Protein samples were supplemented with 1X mPAGE LDS sample buffer (Merck, MP SB) and 50 mM Bolt sample reducing agent (ThermoFisher Scientific, B0009), incubated for 5 min at 95 °C, and separated on a mPAGE 4–12%, Bis-Tris gel (Merck, MP41G12) in MES SDS running buffer (ThermoFisher Scientific, B0002). They were transferred to Immobilon-P PVDF membranes (Merck, IPVH00010) in a Tris/Glycine transfer buffer (25 mM Tris, 192 mM glycine, 20% methanol, pH = 8.3). The blots were blocked with 5% milk for 1 h at room temperature and probed for appropriate primary antibodies: FLAG (mouse; Merck, F3165, 1:5000), TFAM (mouse; Abcam, ab119684, 1:2000), *Drosophila* TFAM (rabbit, custom-made, 1:2000), GFP (chicken; Abcam, ab13970, 1:5000), GFP (rabbit; Antibodies, A290, 1:5000), Poldip2 (rabbit; Proteintech, 15080-1-AP, 1:3000), Poldip2 (rabbit; Abcam, ab181841, 1:2000), PolG1 (rabbit; Abcam, ab128899, 1:2000), *Drosophila* PolG1 (rabbit, custom-made (Yu et al, 2017), 1: 2000) and ClpX (rabbit; ThermoFisher Scientific, PA5-79052, 1:3000), Tubulin (mouse; Developmental Studies Hybridoma Bank, 12G10, 1:10,000), and ATP5A (mouse; Abcam, ab14748, 1:10,000). Blots were visualised by Amersham™ Imager 680 using horseradish peroxidase-conjugated secondary antibodies against chicken (ThermoFisher Scientific, A16054, 1:10,000), mouse (ThermoFisher Scientific, 62-6520, 1:10,000), and rabbit (Novus Biologicals, HAF008, 1:10,000), in combination with Clarity Western ECL substrate (1705061; Bio-Rad Laboratories).

## Pull-down and co-IP assays

Overnight embryos or 200 pairs of testes were collected, lysed in the IP lysis buffer (ThermoFisher Scientific, 87787) supplemented with cOmplete protease inhibitor cocktails (Roche, 11697498001). The samples were incubated on ice for 30 min and then centrifuged at 12,000 × $g$, 4 °C for 15 min. For samples with cross-linking treatment, the following step was applied: the pellet was re-suspended and incubated for 10 min at room temperature in buffer (250 mM sucrose, 20 mM HEPES, 25 mM NaCl, 2 mM EDTA, 0.01% Triton-X) supplemented with 1.5% formaldehyde and 1X protease inhibitor. The sample was then supplemented with 1X glycine and incubated for 5 min at room temperature to stop the cross-linking reaction. The samples (cross-linked or non-cross-linked) were centrifuged at 10,000 × $g$, 4 °C for 5 min and washed three times with PBS. 10% supernatants were aliquoted as input loading, and the rest were incubated overnight at 4 °C with antibodies against FLAG (mouse; Merck, F3165) or GFP (rabbit; Antibodies, A290). The protein-antibody complexes were incubated with Dynabeads protein G beads (Invitrogen, 10004D) at 4 °C for 3 h with constant rotation. After washing the conjugated beads three times with 0.05% Tween20 in PBS, the beads were boiled in 1X LDS PAGE-loading buffer supplemented with protease inhibitor cocktails and 50 mM Bolt sample reducing agent. The immuno-precipitated samples and 10% input were analysed by western

blotting with the following primary antibodies: FLAG (mouse; Merck, F3165, 1:5000), GFP (chicken; Abcam, ab13970, 1:2000; or rabbit; Antibodies, A290, 1:5000), *Drosophila* TFAM (rabbit; custom-made, 1:2000), Poldip2 (rabbit; Abcam, ab181841, 1:2000), *Drosophila* PolG1 (rabbit, custom-made (Yu et al, 2017), 1:2000), and ClpX (rabbit; ThermoFisher Scientific, PA5-79052, 1:2000).

For the pull-down assay, an equal amount of protein for samples and controls was incubated with FLAG antibodies (mouse; Merck, F3165). After eluting from magnetic beads, samples were loaded and separated by mPAGE™ 4–12%, Bis-Tris gel in MES SDS running buffer. The gel was stained with Mass Spectrometry-compatible silver solution according to the manufacturer's protocol (ThermoFisher Scientific, 24600). Unique bands in the Poldip2-PB-FLAG-mCherry lane (Fig. 4A) were excised for mass spectrometry analysis to reveal protein identities.

## mtDNA copy number quantification by qPCR

Genomic DNA was extracted from nine groups of 10 testes or seven groups of soma isolated from 2-d-old male adults as described above. For qPCR assays, 5 µl diluted samples were mixed with 5 µl SensiFast SYBR Green PCR Master Mix (Bioline, 98020) with 0.5 µM of each primer: HEXA-F/HEXA-R (nuclear primers) and mt361F/mt409R (mtDNA primers). qPCR was executed on the QuantStudio 3 (ThermoFisher Scientific) with the following conditions: 95 °C for 10 min, 40 cycles of 95 °C for 15 s and 54 °C for 15 s. The relative quantification ($\Delta C_T$) method was employed to compare mtDNA levels relative to nuclear DNA levels.

## ChIP-qPCR

Overnight embryos were collected, dechlorinated in 50% bleach and dounced in ice-cold homogenisation buffer (250 mM sucrose, 10 mM Tris-HCl, 10 mM EDTA pH = 8.8, 1% BSA, and 1X protease inhibitor). After centrifugation at 100 × $g$, 4 °C for 5 min, the supernatant was transferred to a new pre-chilled tube and centrifuged at 10,000 × $g$, 4 °C for 5 min. The pellet was re-suspended and incubated for 10 min at room temperature in cross-linking buffer (250 mM sucrose, 20 mM HEPES, 25 mM NaCl, 2 mM EDTA, 0.01% Triton-X) with freshly added 1.5% formalde-hyde and 1X protease inhibitor. The sample was then supplemented with 1X glycine and incubated for 5 min at room temperature to stop the cross-linking reaction. The samples were centrifuged at 10,000 × $g$, 4 °C for 5 min and washed three times with PBS. SimpleChIP® Plus Kit protocol (Cell Signaling Technology, 9005) was followed for the rest of the procedures, except for centrifuga-tion steps (10,000 × $g$, 4 °C for 5 min). In brief, the DNA-protein complexes were digested with micrococcal nucleases. Then, an equal amount of the sheared DNA-protein complexes of each sample was aliquoted for 2% Input, and the rest were immuno-precipitated with FLAG (mouse; Merck, F3165), GFP (rabbit; Antibodies, A290), and appropriate IgG control (mouse, Merck, I5381; or rabbit, Cell Signaling Technology, 2729). After elution and DNA purification steps, qPCR was used to identify DNA-binding regions and quantify DNA fold-enrichment in each IP sample compared to their inputs. qPCR was performed in the same condition as mentioned above, except there were 6 pairs of mitochondrial primers and one pair of nuclear primer (HEXA)

(Appendix Table S2). mtDNA content in each sample was calculated from $2^{\Delta CT}$, when $\Delta CT = C_T \text{ nDNA} - C_T \text{ mtDNA}$.

## Proteinase K protection assay

The proteinase K protection assay was performed as previously described (Klucnika et al, 2023). Briefly, HEK293 cells or *Drosophila* tissues were dounced in homogenisation buffer (210 mM mannitol, 70 mM sucrose, 10 mM HEPES, 1 mM EDTA, pH 7.4) plus 1% protease inhibitor cocktail (Merck, 11836170001). The cell debris and nucleus were removed by centrifuging at $500 \times g$ for 5 min at 4 °C and then $1000 \times g$ for 5 min at 4 °C. The mitochondria were pelleted by centrifuging at $5000 \times g$ for 10 min at 4 °C and washed at $5000 \times g$ for 10 min at 4 °C in homogenisation buffers without protease inhibitors. The mitochondrial pellet was then suspended in homogenisation buffers and the protein concentration was measured using the Qubit protein assay kit (ThermoFisher Scientific, Q33211). For the proteinase K protection assay, 50 mg of mitochondrial proteins were pelleted by centrifugation at $5000 \times g$ for 10 min at 4 °C and re-suspended in 500 μl of homogenisation buffer, mitoplast/swelling buffer (10 mM HEPES, pH 7.4), or solubilising buffer (homogenisation buffer with 1% Triton X-100), and incubated on ice for 15 min. The mitoplast/swelling sample was pipetted up and down 15 times to disrupt the mitochondrial outer membrane. Proteinase K (New England Biolabs, P8107S) was then added to the samples to a final concentration of 4 U/ml, and samples were incubated on ice for 20 min. To terminate the reaction, PMSF was added to all samples to a final concentration of 2 mM followed by 5 min incubation on ice. The resulting proteins were then precipitated by 12.5% TCA, washed with cold acetone, re-suspended in 100 μl of 1X LDS sample buffer, and boiled for 5 min. Finally, 20 μl of samples were analysed by Western blot. Tom20 (rabbit; Abcam, ab186735, 1:2000), Tim50 (rabbit; Proteintech, 22229-1-AP, 1:2000) and TFAM (mouse; Abcam, ab119684, 1:2000) were probed as markers for different compartments of human HEK293 mitochondria. Porin (rabbit; Merck, PC546, 1:1000), Opa1-HA (HA antibody: rabbit; Cell Signaling Technology, 3724S, 1:2000), and ATP5A (mouse; Abcam, ab14748, 1:10,000) were probed as markers for different compartments of fly mitochondria.

## Measurement of paternal mtDNA copy number in progeny by ddPCR

Genomic DNA was extracted from groups of 10 adult progeny (each group contains 10 flies) as mentioned above. ddPCR was conducted with QIAcuity EG PCR Kit (QIAGEN, 250113) on QIAcuity Nanoplate 8.5k or 26k plates. The DNA samples were first diluted to the target concentration (1:200–1:10,000), and then mixed with the master mix as suggested by the protocol. The ddPCR was performed on QIAcuity One, 5 plex System (QIAGEN, 911022) using primers (listed in Appendix Table S2) and the following thermocycles: initial heat activation 95 °C for 2 min, 40 cycles of 95 °C 15 s, 58 °C 15 s, 72 °C 15 s, followed by 5 min of cooling down at 4 °C. Results were analysed by QIAcuity Software Suite Version 1.2.

## RNAseq and data analysis

The RNA was extracted by TRIzol (ThermoFisher Scientific, 15596026) following the manufacturer's instructions. In brief,

groups of 50 testes for different genotypes were ground in 750 μl of TRIzol reagent and incubated at room temperature for 10 min. Phenol was removed from samples by multiple rounds of chloroform extraction. RNA from the supernatant was precipitated by adding 0.5X isopropanol and washed once with 70% ethanol. The extracted RNA was then treated with RNase-free DNase I (New England Biolabs, M0303) for 30 min at 37 °C to remove DNA followed by heat-inactivating DNase I by incubating 10 min at 65 °C upon adding 1 μl of 50 mM EDTA.

Approximately 500 ng of total RNA was used for library prep. Following DNase I digestion, long RNA (>200 nt) was isolated using Zymo Clean and Concentrator columns (Zymo, R1019). Subsequently, rRNA was removed using the NEBNext mRNA Magnetic Isolation Module (New England Biolabs, E7490L). With the purified RNA, cDNA libraries were prepared by employing the NEBNext Ultra Directional RNA Library Prep Kit (New England Biolabs, E7760) and double-stranded cDNA was purified using AMPure XP Beads (Beckman Coulter, A63881). The prepared RNA libraries were sequenced on a NovaSeq600 in paired-end mode.

RNAseq data was processed using nf-core/rnaseq pipeline v3.8.1 (Love et al, 2014) implemented in nextflow (version 21.10.6). The custom *Drosophila* genome and GTF file obtained from UCSC were used as a reference genome. Other parameters for the nextflow pipeline were kept as default. In brief, the nf-core/rnaseq pipelines involve a quality check of raw reads using FastQC, trimming low-quality and adaptors containing reads using TrimGalore, alignment of good quality reads using STAR, deduplication by PICARD tool and quantification by Salmon. All the output reports are assessed by Multiqc. The raw counts were subjective to variance stabilising transformation using functions implemented in DESeq2 v2.36.0 (Patel et al, 2022).

## Statistics analysis and reproducibility

Statistical analyses were performed and plotted using Prism 9 (GraphPad). Data are presented as means ± SD. Comparisons of different samples were performed using unpaired Student's t-test or one-way ANOVA with Tukey's test as specified in figure legends. For the Student's t-test, data distribution was assumed to be normal, but this was not formally tested. Animals or samples were randomised and exposed to the same environment. The mtDNA quantification by qPCR and ddPCR assay, and fertility tests were conducted in a blind manner, and the identity of samples was disclosed only after the completion of data analysis. No data were excluded from analyses.

# Data availability

All experimental data are presented in the main text or supplementary materials. DNA and RNA sequencing data has been deposited in ArrayExpress (accessions E-MTAB-14100 and E-MTAB-14104). An in-house bioinformatics pipeline was developed to identify mutations in sequenced EMS lines. The code and instructions for running the analysis have been uploaded on the dmvar GitHub page (https://github.com/adamjamesreid/dmvar).

The source data of this paper are collected in the following database record: biostudies:S-SCDT-10_1038-S44318-025-00378-4.

## Peer review information

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

## Acknowledgements

We thank the Imaging, Sequencing, and Mass Spectrometry Facilities at the Gurdon Institute and the University of Birmingham for their support. We also thank Professor Patrick O'Farrell at the University of California San Francisco for sharing their custom-made antibodies against *Drosophila* PolG1. This work was funded by Wellcome Sir Henry Dale Fellowship 202269/A/16/Z, ERC Starting Grant 803852 and Philip Leverhulme Prize PLP-2020-063 to Hansong Ma. The Gurdon Institute Core Facility was funded by Wellcome Trust grant 203144 and Cancer Research UK grant C6946/A24843.

## Author contributions

**Ziming Wang**: Conceptualization; Data curation; Formal analysis; Validation; Investigation; Visualization; Methodology; Writing—original draft; Writing—review and editing. **Tirawit Meerod**: Formal analysis; Validation; Investigation; Visualization; Methodology; Writing—review and editing. **Nuria Cortes-Silva**: Conceptualization; Formal analysis; Validation; Investigation; Methodology; Writing—review and editing. **Ason C-Y Chiang**: Conceptualization; Formal analysis; Supervision; Validation; Investigation; Visualization; Methodology; Writing—original draft; Writing—review and editing. **Ziyan Nie**: Investigation; Writing—review and editing. **Ying Di**: Investigation; Writing—review and editing. **Peiqiang Mu**: Investigation; Writing—review and editing. **Ankit Verma**: Formal analysis. **Adam James Reid**: Formal analysis; Methodology; Writing—review and editing. **Hansong Ma**: Conceptualization; Supervision; Funding acquisition; Validation; Investigation; Methodology; Writing—original draft; Project administration; Writing—review and editing.

Source data underlying figure panels in this paper may have individual authorship assigned. Where available, figure panel/source data authorship is listed in the following database record: biostudies:S-SCDT-10_1038-S44318-025-00378-4.

## Disclosure and competing interests statement

The authors declare no competing interests.

# Expanded View Figures

**Figure EV1.   The EMS screen used to identify mutant lines retaining paternal mtDNA in mature sperm.**

(**A**) The cross scheme for establishing ~10,000 EMS lines carrying random mutations on the 2nd or 3rd chromosome of *D. melanogaster*. The following genotypes were used as the parental lines for the EMS mutagenesis: 1) *FRT40A* (Chr. 2L); *sqh-mitoYFP*, 2) *FRTG13* (Chr. 2R); *sqh-mitoYFP*, 3) *ubi-mtSSB-GFP*; *FRT2A* (Chr. 3L), and 4) *ubi-mtSSB-RFP*; *FRT82B* (Chr. 3R). Fly lines with balancer chromosomes used for subsequent crosses to establish individual EMS lines are as follows: 1) *Kr[If]/CyO*; *sqh-mitoYFP* (Chr. 2L and Chr. 2R), 2) *ubi-mtSSB-GFP*; *MKRS/TM6B* (Chr. 3L), and 3) *ubi-mtSSB-RFP*; *MKRS/TM6B* (Chr. 3R). (**B**) Representative images showing DAPI, dsDNA, mtSSB-GFP and PicoGreen signals in spermatocytes and spermatids. Scale bars: 10 μm.

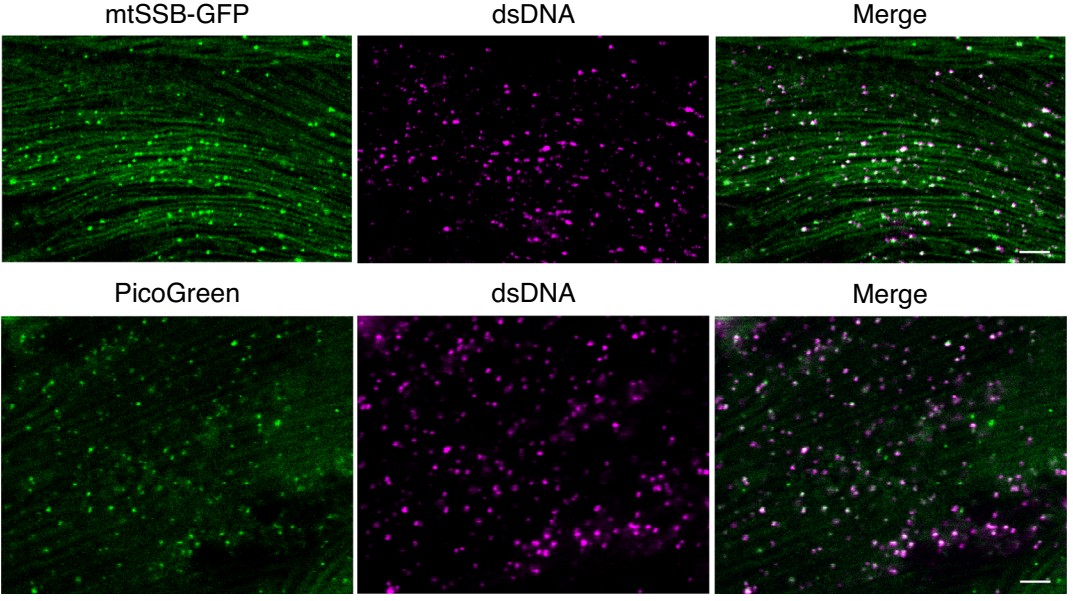

**A**

Males with a FRT site
(*FRT40A, G13, 2A*, or *82B*) ♂

EMS mutagenesis (25 mM)

Group mating  ♂ ×  Balancer stock ♀

Single mating  ♂ ×  Balancer stock ♀

Self mating  ♂  ♀   ~10,000 EMS lines established,
4621 were homozygous viable

**B**

Spermatocytes

| mtSSB-GFP | dsDNA | DAPI | Merge |

Spermatids at mid-elongation stage

| mtSSB-GFP | dsDNA | Merge |

| PicoGreen | dsDNA | Merge |

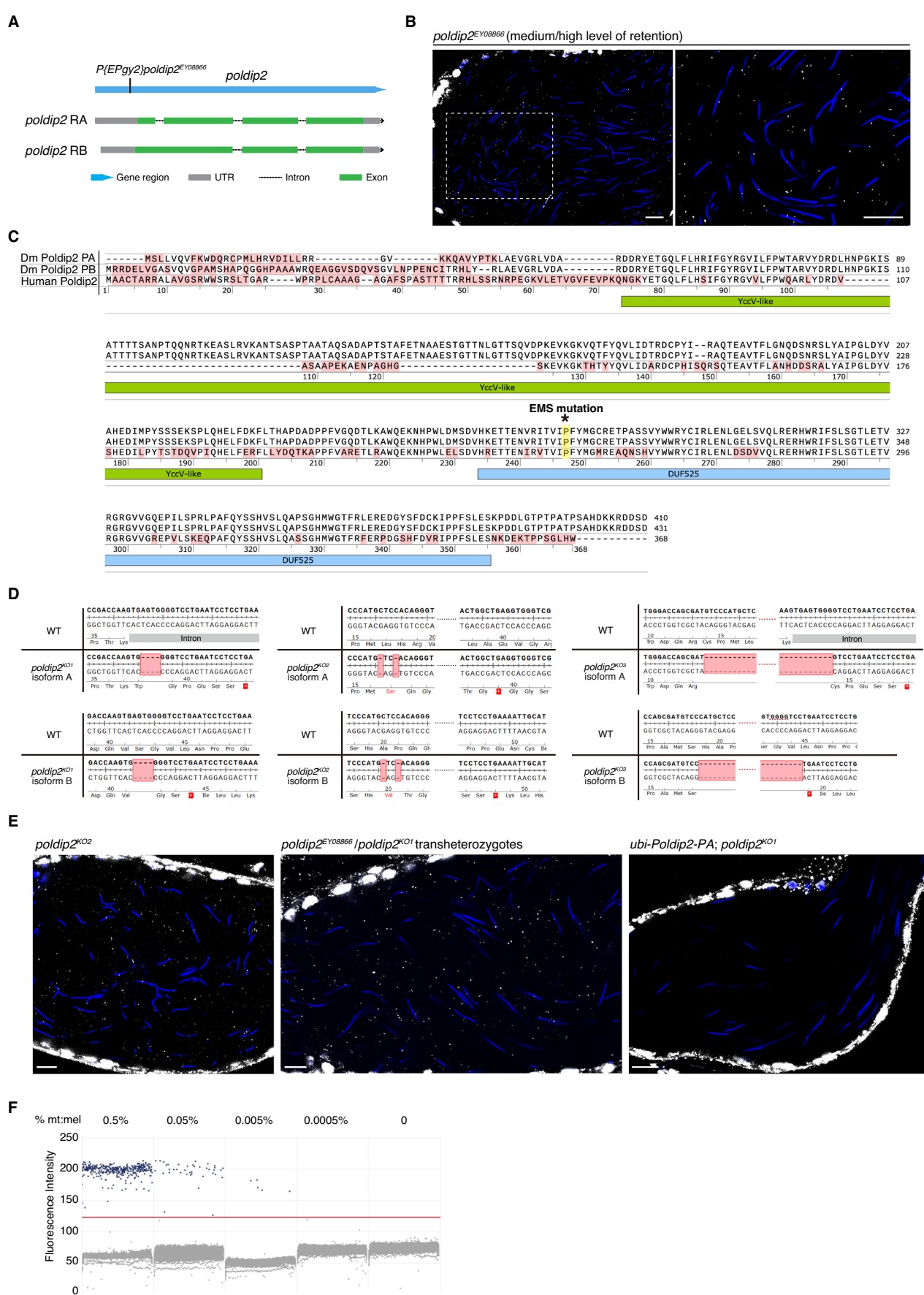

◀    **Figure EV2.   Poldip2 is required for mtDNA removal in *Drosophila* spermatids.**

(A) Schematic of the P-element insertion site for the *poldip2^{EY08866}* line. (B) A representative image showing the level of mtDNA retention in seminal vesicles of *poldip2^{EY08866}* mutant. The samples were stained with anti-dsDNA antibodies (white, mtDNA) and DAPI (blue, nuclear DNA). Scale bars: 10 μm. (C) The alignment of the Poldip2 sequence of two fly isoforms and the human protein, with the positions of YccV-like and DUF525 domains annotated. The *EMS-23* carries a point mutation that converts the highly conserved proline to serine in the DUF525 domain (highlighted in yellow with an asterisk mark). (D) Sequences of *poldip2* mutants generated in this study. (E) Representative images with the level of mtDNA retention in seminal vesicles of *poldip^{KO2}*, *poldip2^{EY08866}*/*poldip2^{KO1}* transheterozygotes, and *ubi-Poldip2-PA-FLAG-mCherry*; *poldip2^{KO1}* flies. The samples were stained with anti-dsDNA antibodies (white, mtDNA) and DAPI (blue, nuclear DNA). Scale bars: 10 μm. (F) Plots illustrating the detection power of our ddPCR assay. *D. melanogaster* mtDNA was mixed with *D. yakuba* mtDNA to generate samples with 0.5%, 0.05%, 0.005% and 0.0005% of mt:mel before the run. The red line marks the threshold, above which droplets were considered positive for mt:mel (i.e. paternal mtDNA).

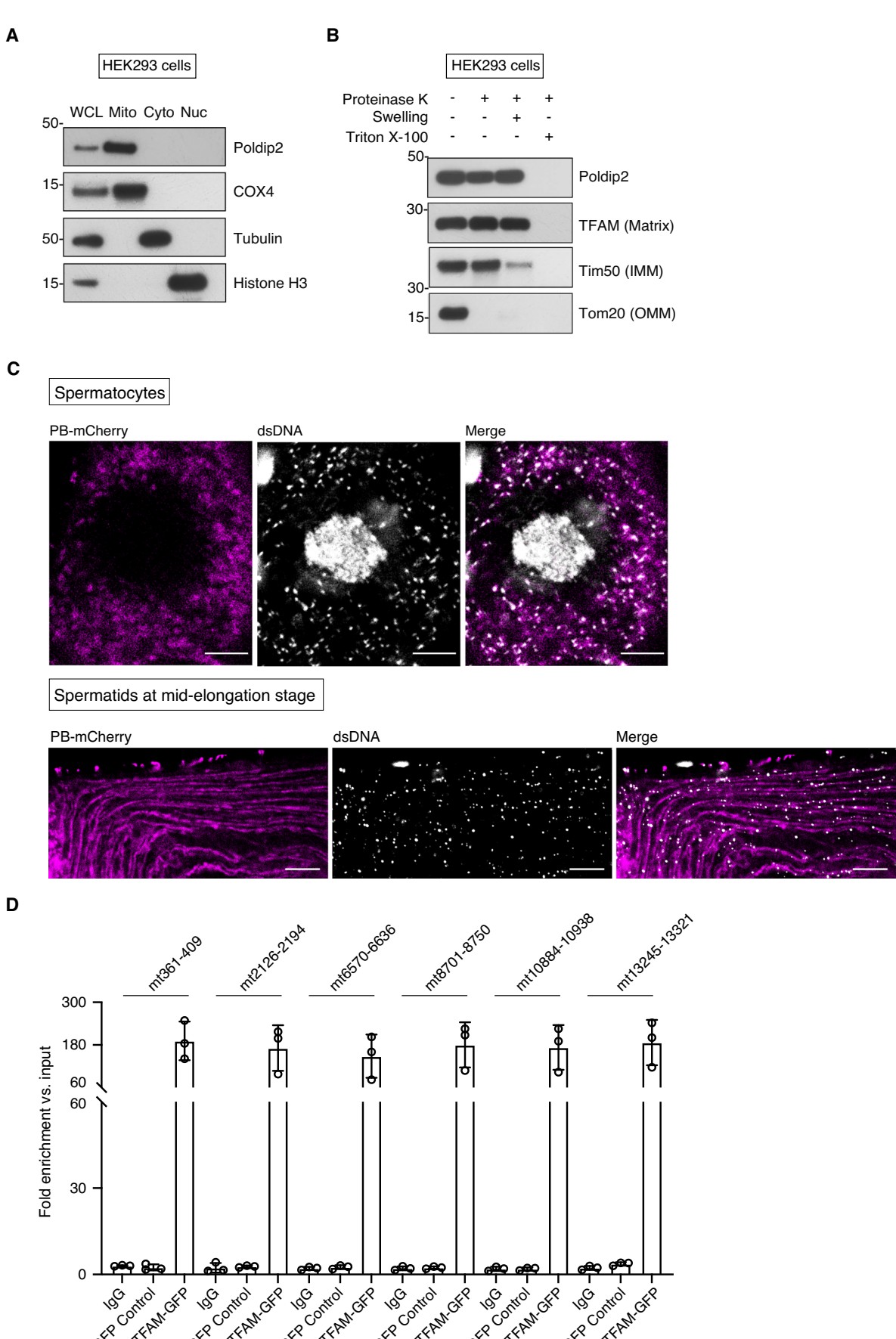

**Figure EV3. Poldip2 is a mitochondrial matrix protein.**

(A) Immunoblots of whole-cell lysate (WCL), mitochondrial (Mito, anti-COX4), cytoplasmic (Cyto, anti-Tubulin) and nuclear (Nuc, anti-Histone H3) fractions of HEK293 cells to reveal the subcellar enrichment of Poldip2. (B) Immunoblots of proteinase K protection assay in HEK293 cells. TFAM, Tim50, and Tom20 were blotted as the mitochondrial matrix, inner membrane (IMM) and outer membrane (OMM) marker, respectively. (C) Confocal images showing Poldip2-mCherry signals (magenta) and dsDNA foci (white) in spermatocytes and spermatids of *ubi-Poldip2-PB-FLAG-mCherry* flies. Scale bars: 5 μm. (D) ChIP-qPCR measuring the mtDNA enrichment levels with TFAM-GFP immunoprecipitation in flies ($n = 3$ biological replicates). Embryos were used to obtain sufficient materials. Data were normalised to input DNA. IgG control: immunoprecipitating TFAM-GFP samples with IgG; GFP control: immunoprecipitating wild-type samples (i.e. no GFP expression) with anti-GFP antibodies to control for unspecific bindings between anti-GFP antibodies and mtDNA during the assay; TFAM GFP: immunoprecipitating TFAM with GFP antibodies. Three independent ChIP assays were performed for TFAM-GFP, and the mtDNA enrichment level was measured by qPCR using six pairs of primers binding to different regions of mtDNA. Data: mean ± SD.

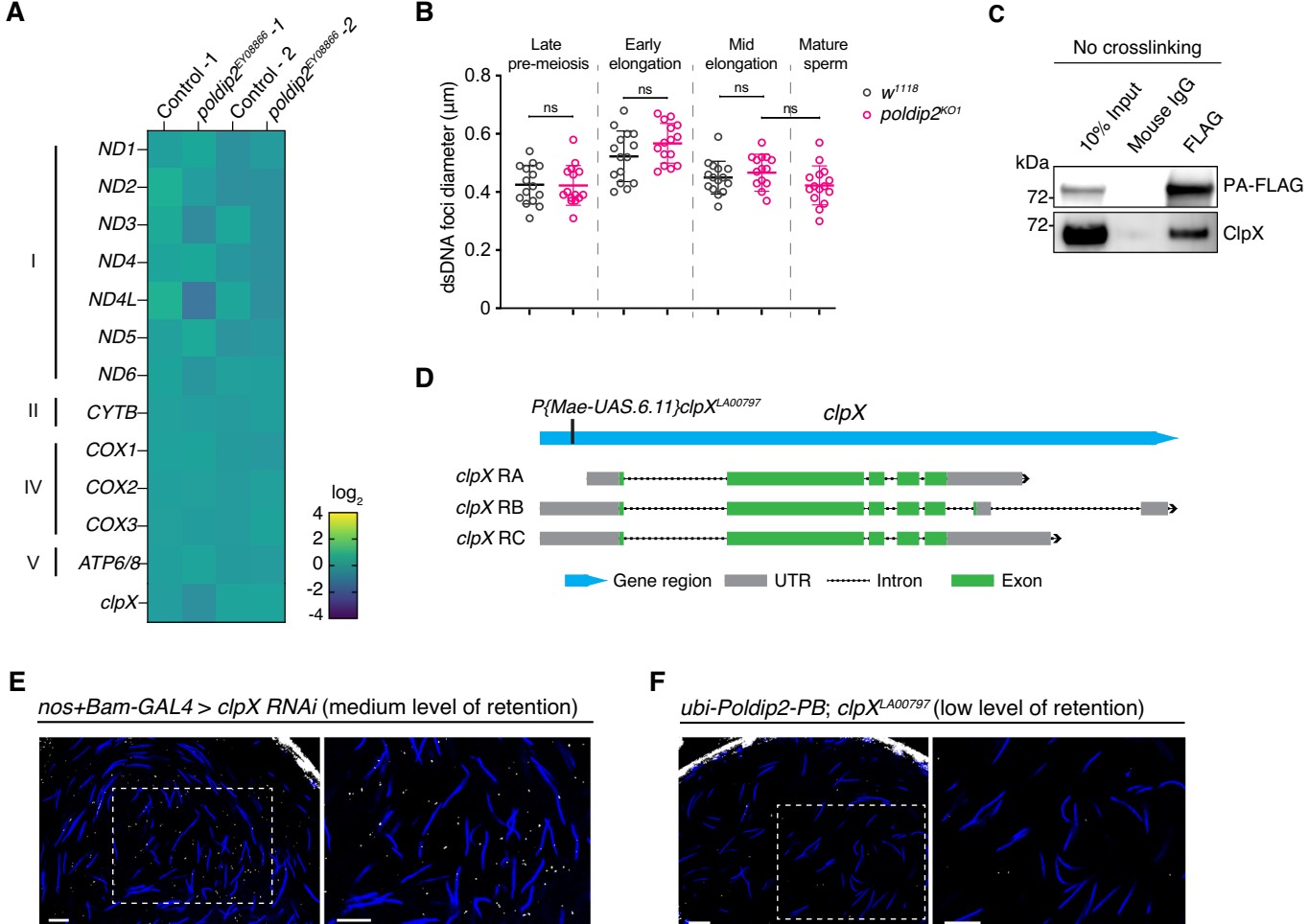

**Figure EV4. ClpX and Poldip2 co-regulate mtDNA elimination in *Drosophila* spermatids.**

(A) A heatmap with mRNA levels of mtDNA genes in control and *poldip2^EY08866^* testes. (B) Diameters of mtDNA nucleoid at different stages of spermatogenesis ($n = 15$ biological replicates). Data: mean ± SD, unpaired Student's t-test, $P = 0.9145$, $0.1285$, $0.4542$ and $0.0795$, respectively. (C) Immunoblots of co-IP using anti-FLAG antibodies. Testes isolated from *ubi-Poldip2-PA-FLAG-mCherry* flies were used, and ClpX was blotted. (D) Schematic of the P-element insertion site for the *clpX^LA00797^* line. (E) A representative image of seminal vesicles isolated from *clpX* RNAi; *nos+Bam-GAL4* flies stained with anti-dsDNA antibodies (white) and DAPI (blue). Scale bars: 10 μm. (F) A representative image of seminal vesicles isolated from *ubi-Poldip2-PB*; *clpX^LA00797^* flies stained with anti-dsDNA antibodies (white) and DAPI (blue). Scale bars: 10 μm.

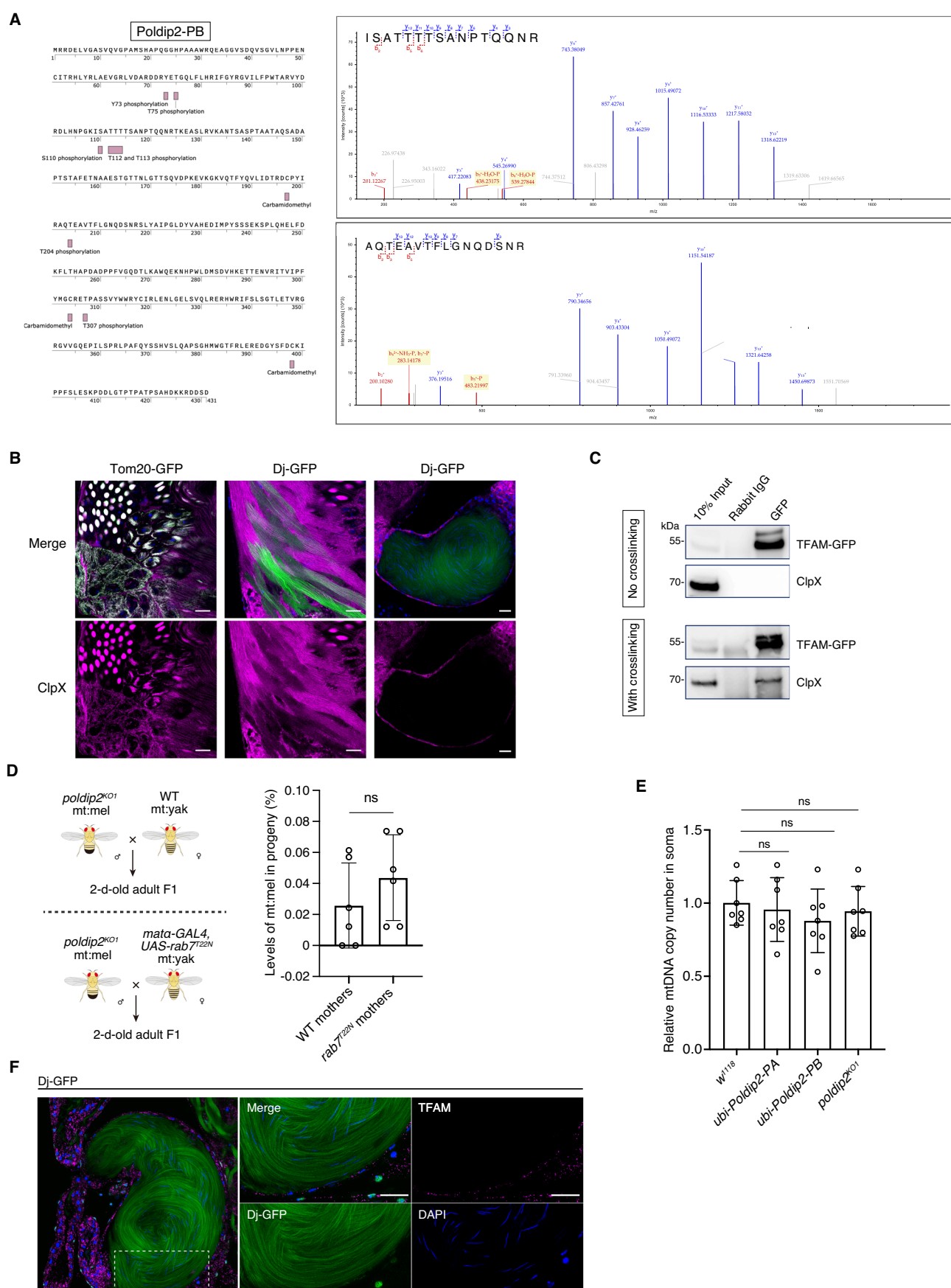

◄   **Figure EV5.   The role of Poldip2 and ClpX in regulating mtDNA dynamics during *Drosophila* spermatogenesis.**

(A) Post-translational modification sites of Poldip2-PB identified by mass spectrometry analyses. The left panel notes all the phosphorylation and carbamidomethylation sites based on the MS1 spectra. The right panel is the two MS2 spectra plots for T112/113 and T204 phosphorylation sites. (B) Representative images showing ClpX expression in spermatocytes, spermatids and mature sperm. ClpX (magenta) was visualised by immunostaining, whereas *Dj-GFP* and *ubi-Tom20-GFP* (green) were used to visualise mitochondria. Samples were also stained with DAPI (blue). Scale bars: 20 μm. (C) Immunoblots of co-IP with anti-GFP antibodies to probe interactions between TFAM or ClpX in testes with or without cross-linking. (D) Percentages of paternal mtDNA in 2-d-old adult progeny of *poldip2^{KO1}* males when crossed to wild-type females or females expressing Rab7$^{T22N}$ ($n = 6$ crosses). Data: mean ± SD, unpaired Student's t-test, $P = 0.2859$. (E) The relative mtDNA copy number in soma measured by qPCR for *w^{1118}*, *ubi-Poldip2-PA*, *ubi-Poldip2-PB* and *poldip2^{KO1}* males. The number was normalised to *w^{1118}* samples ($n = 7$). Data: mean ± SD, One-Way ANOVA, $P = 0.9438$, $0.4985$ and $0.8983$, respectively. (F) Representative images showing TFAM localisation in seminal vesicles isolated from *Dj-GFP* flies. TFAM (magenta) was visualised by immunostaining, whereas *Dj-GFP* (green) was used to visualise mitochondria. Samples were also stained with DAPI (blue). Scale bars: 20 μm.

