## [Peer Review File · The EMBO Journal]

Poldip2 promotes mtDNA elimination during *Drosophila* spermatogenesis to ensure maternal inheritance

Ziming Wang, Tirawit Meerod, Nuria Cortes-Silva, Chieh-Yin Chiang, Ziyang Nie, Ying Di, Peiqiang Mu, Ankit Verma, Adam Reid and Hansong Ma

Corresponding author: Hansong Ma (h.ma.6@bham.ac.uk)

Review Timeline:

Submission Date:	3rd Jun 24
Editorial Decision:	5th Aug 24
Revision Received:	2nd Oct 24
Editorial Decision:	6th Dec 24
Revision Received:	8th Dec 24
Accepted:	24th Jan 25

Editor: Ieva Gailite

Transaction Report:

Dear Dr. Ma,

Thank you for submitting your manuscript for consideration by the EMBO Journal. I sincerely apologise for the protracted assessment process due to delays in referee report submission. We have now received comments from three reviewers, which are included below for your information.

As you can see, all reviewers are generally positive in their assessment and find the proposed role of Poldip2 novel and of interest to the readership of our journal. However, they also indicate several concerns that would need to be addressed before they can support publication. From my side, I find the raised points generally reasonable. I would therefore invite you to address these remaining comments in a revised manuscript. I think that it would be useful to discuss the revision in more detail via email or phone/videoconferencing - please let me know which option you prefer.

We generally allow three months as standard revision time. Should you foresee a problem in meeting this deadline, please let us know in advance to discuss an extension. As a matter of policy, competing manuscripts published during this period will not negatively impact on our assessment of the conceptual advance presented by your study. However, please contact me as soon as possible upon publication of any related work to discuss the appropriate course of action.

When preparing your letter of response to the referees' comments, please bear in mind that this will form part of the Review Process File and will therefore be available online to the community. For more details on our Transparent Editorial Process, please visit our website: <https://www.embopress.org/page/journal/14602075/authorguide#transparentprocess>. Please also see the attached instructions for further guidelines on preparation of the revised manuscript.

Please feel free to contact me if have any further questions regarding the revision. Thank you for the opportunity to consider your work for publication, and I look forward to discussing your revision with you.

With best wishes,

leva

leva Gailite, PhD
Senior Scientific Editor
The EMBO Journal
Meyershofstrasse 1
D-69117 Heidelberg
Tel: +4962218891309
i.gailite@embojournal.org

- a point-by-point response to the referees' comments, with a detailed description of the changes made (as a word file).
- a word file of the manuscript text.
- individual production quality figure files (one file per figure)
- a complete author checklist, which you can download from our author guidelines (<https://www.embopress.org/page/journal/14602075/authorguide>).

- Expanded View files (replacing Supplementary Information)

- a Reagents and Tools Table as part of the Methods section, which can be downloaded from our author guidelines

(<https://www.embopress.org/page/journal/14602075/authorguide#structuredmethods>)

We realize that it is difficult to revise to a specific deadline. In the interest of protecting the conceptual advance provided by the work, we recommend a revision within 3 months (3rd Nov 2024). Please discuss the revision progress ahead of this time with the editor if you require more time to complete the revisions.

Referee #1:

The study by Wang et al. used an unbiased genetic screen to identify new genes required for mtDNA elimination during spermatogenesis. Wang et al. focus on Poldip2 (also identified by Chen et al.) and show that is required for complete paternal mtDNA elimination. Similar to Chen et al., Wang et al. show that Poldip2 mutants are semi sterile. However, Wang et al. additionally show that Poldip2 mutant fathers can pass their mtDNA to many (or perhaps all) of their offspring. Why mtDNA is almost always inherited from a single parent is an unresolved and broadly interesting question in genetics, that will be interesting to the readers of EMBO. While Wang et al. do not resolve this question, their discovery that Poldip2 fathers can transmit mtDNA to their offspring is a very important first step that will promote future research on the topic. The following discoveries in Wang et al. are important and are well supported by the data:

- 1) Poldip2 is required for complete elimination of mtDNA during late spermatogenesis
- 2) Poldip2 is a mitochondrial matrix protein that can associate with mtDNA
- 3) Poldip2 mutants are semi sterile
- 4) Poldip2 mutant fathers transmit mtDNA to their offspring
- 5) Poldip2 physically interacts with ClpX
- 6) Similar to Poldip2, ClpX is required for complete mtDNA elimination during late spermatogenesis.

There are no major problems with this paper and in my opinion, it is suitable for publication in your journal with only minor revisions.

Minor Criticisms:

1) The authors' EMS screen seems to be a highly successful method for studying mtDNA elimination during spermatogenesis. The authors found 47 mutant lines with high mtDNA retention but focused on one line, #23, to characterize in this paper. Do the other lines harbor alleles of Poldip2, or the other two genes previously reported to be involved in mtDNA elimination during spermatogenesis: EndoG and Tamas? The strains that the authors mutagenized should also allow them to roughly map the causative allele to one of 4 chromosome arms. Did the authors do this mapping? If so, did they perform complementation tests with mutants that mapped to the same arm? From their screen, can the authors estimate how many genes contribute to mtDNA elimination during spermatogenesis? Or can they at least comment on how many new genes they found?

2) The paragraph about the mt:ND2del1 assay to measure paternal leakage doesn't refer to any data presented as figures and should be removed.

3) The Dmel x Dyak assay to detect paternal leakage is very elegant. To strengthen their conclusion that Poldip2 but not WT fathers transmit mtDNA to their adult offspring, the authors should report the raw data for the control WT cross- particularly because Chen et al. did not detect paternal transmission. The authors should also consider reporting the sensitivity threshold of their assay. For example, could their assay detect 0.0001% Dmel in Dyak? From the measured sensitivity, do they think they could detect a single mtDNA molecule inherited through a sperm?

4) Similarly, the authors state, "we anticipate the initial leaked amount in embryos to be much lower than the ddPCR values." The authors should expand on this interpretation to help the reader follow. How many copies of mtDNA are in an egg and what is the range of copy numbers (and paternal mtDNA percentages) that could be transmitted by sperm? If a sperm has at most

200 copies and an egg has about 2 million copies, one might observe initial paternal transmission to be anywhere from 0 to 0.01%.

5) Regarding the TFAM ChIP qPCR experiments, the authors state "This significantly higher mtDNA enrichment compared to POLDIP2 is anticipated, as TFAM is known to bind broadly across the mitochondrial genome with high affinity and quantities. The difference in mtDNA enrichment between TFAM and POLDIP2 ChIP assay likely arises from differences in their mtDNA binding affinities and quantities." An alternative explanation is that different antibodies have different affinities for target proteins and sources of background. The authors should just remove the above statement and the comparison between Poldip2 and TFAM binding to mtDNA because it is not essential or important for the paper.

6) In the next paragraph, the authors state, "The observed enrichment of mtDNA with POLDIP2 immunoprecipitation is unlikely attributed to interactions between POLDIP2 and other mtDNA binding proteins, as our co-immunoprecipitation (co-IP) experiments detected weak/no interaction between POLDIP2 and TFAM, mtSSB, or mtDNA polymerase POLG1 in fly tissues and HEK293 cells (Fig 3G, S3D)." These co-IP experiments have much less sensitivity than ChIP. Failing to detect a protein-protein interaction in co-IP does not mean that this interaction is not part of a protein-protein-DNA interaction in ChIP. The authors should just remove the above statement.

7) Chen et al. found that Poldip2 expression coincided with mtDNA elimination during late spermatogenesis. An interesting question is whether Poldip2 expression induces mtDNA elimination. The authors created ubi-Poldip2 transgenic fly lines, which theoretically constitutively express Poldip2. Did these lines constitutively express Poldip2 and did this result in reduced mtDNA in any of the tissues expressing Poldip2?

8) The connection between Poldip2 and ClpX is interesting because both genes physically interact and have a similar mtDNA elimination defect. However, the author's model where Poldip2 recruits ClpX to mtDNA to help remove mtDNA-interacting proteins is quite speculative and belongs in the discussion section.

9) In the discussion section, the authors speculate, "POLDIP2 could undergo continuous degradation via post-translational regulation in earlier stages to prevent it from interacting with mtDNA and trigger its degradation." The authors should be able to test this by looking at their ubi-Poldip2 transgene, which is presumably constitutively expressed.

Referee #2:

Using forward genetic screening, Wang et al. identified POLDIP2 as a factor regulating mtDNA clearance during spermatogenesis in *Drosophila melanogaster*. They showed that POLDIP2 is a mitochondrial matrix protein and binds to mtDNA in ChIP assay. Using mass spec analysis, they further identified CLPX, a subunit of matrix protease complex CLPXP, as a POLDIP2 binding protein. The ClpX hypomorphic mutant also showed a defect in mtDNA clearance in sperm, and this phenotype was rescued by Poldip2 overexpression. These results suggest that POLDIP2 and CLPX function together in mtDNA clearance.

Overall, this work is well organized and provides important mechanistic insights into mtDNA clearance in fly sperm. Although many of the conclusions are well supported by the data, some important points should be addressed before publication.

Major comments

1. In Fig.2, the authors showed that the mtDNA signals remain in the seminal vesicles of the Poldip2 mutant. It would be better to include some quantification of remaining mtDNA nucleoids in each sperm.
2. We cannot see whether the organization of mtDNA nucleoids is affected by the Poldip2 mutation in Fig. 2. Please mention this point (size, number, and position, for example). It would also be informative to include enlarged images of the wild-type and mutant spermatids at various stages to compare morphology and number of nucleoids.
3. The authors showed that mCherry-tagged POLDIP2 PA and PB colocalized with mitochondrial markers. It should be mentioned whether these tagged transgenes are functional. I also wonder whether POLDIP2 localizes to mtDNA nucleoids or localizes uniformly in the matrix. Enlarged images of Fig. 3C and D would be helpful to address this point.
4. Is the amount of TFAM or other nucleoid proteins (e. g. mtSSB, POLG1) in the testis affected by the Poldip2 and ClpX mutations? It would be also important to examine the protein dynamics of ClpX during spermatogenesis. Does ClpX localize to nucleoid in this process?
5. The Poldip2 mutant males showed reduced fertility. Please clarify the morphology and number of mature sperm in the Poldip2 mutant.
6. In Fig. 4D, the authors showed that the phenotype of the ClpX mutant was rescued by Poldip2 overexpression by using qPCR

of mtDNA in the testis. Because the testis also includes somatic cells, it is better to confirm this result by mtDNA staining of sperm in seminal vesicles.

7. Poldip2 was originally identified as a human polymerase P50-interacting protein and has been proposed to be involved in nuclear genome replication and repair. In this study, the authors seem to agree with the conclusion proposed by Chen et al that Poldip2 functions as a DNA exonuclease. Does Poldip2 contain any conserved DNA exonuclease catalytic domain? If not, the authors should confirm the DNA exonuclease activity of Poldip2 in vitro by themselves. It would be essential to examine DNA-binding defective mutant (Yvvc domain mutations) and hypomorphic mutants identified by the authors as well as wild-type proteins.

8. It has been reported that rab7 and atg7 are involved in the degradation of paternal mitochondria in embryos. Do these mutations (rab7 or atg7) enhance the paternal mtDNA leakage phenotype of Poldip2 and ClpX mutants? Do Poldip2 and ClpX double mutants show enhanced paternal mtDNA leakage phenotype?

Minor comments

1. It would be informative to mention the expression pattern of CLPX and the progeny size of the mutant.

2. Fig. 1C, Fig. 3E-G, Fig. 4A-C, F, Fig. S4B, C

Please indicate which tissue was used in the legend of each Figure (or directly in the figures).

3. In Fig. 2F, paternal leakage of mtDNA was addressed by ddPCR. Please indicate which stage of progeny was used in the main text and Figure 2 legend (According to Methods, adult progeny was used).

4. Please indicate the position of P-element insertion for Poldip2 EY08866 and CLPX LA00797.

5. p6 line 2

"On average, each male adult carries $\sim 10^8$ copies of mtDNA."

Please include references.

6. Fig. S2D right panel

The genotype is indicated as ubi-Poldip2-PB-FLAG-mCherry; Poldip2KO in the legend but ubi-Poldip2-PA; Poldip2KO in Figure.

Referee #3:

Summary and significance:

Wang et al. report the finding of Poldip2 through a forward mutagenesis screen in *Drosophila melanogaster*, identifying 47 mutants that exhibited retention of mtDNA in mature sperm. Using deficiency mapping and sequencing, the authors mapped one of the mutants to the Poldip2. The authors use proteinase K protection assays, ChIP, and fluorescence imaging to characterize Poldip2 as a mitochondrial matrix protein that can bind to mtDNA. Poldip2 mutant males are able to transmit their mtDNA to their offspring if mated with *D. melanogaster* females carrying exclusively *D. yakuba* mtDNA, which is less competitive in the background of a *D. melanogaster* nuclear genome. They show that Poldip2 expression coincides with mtDNA degradation in the testis, and show that Poldip2 interacts and stabilizes CLPX, a protease. A CLPX hypomorphic allele that leads to a substantial reduction in protein levels phenocopies the Poldip2 mutant, which can be rescued by overexpression of a Poldip2 isoform. The authors suggest that CLPX contributes to mtDNA clearance in testes, presumably in a co-regulative process together with Poldip2.

We consider this study an interesting and important body of work. The experiments are well-designed, and the data is presented clearly. The identification of factors contributing to mtDNA clearance in *Drosophila* sperm adds an important piece to the puzzle of safeguarding maternal inheritance of mitochondrial DNA.

Specific concerns:

The manuscript would benefit from a clearer description of the nature of the interaction between Poldip2 and CLPX as well as the potential function of CLPX during this process. The proposed model in Fig 4 hints towards a possible influence on TFAM levels bound to mtDNA, but this is speculative as there is no data in the paper that speaks specifically to the impact on TFAM levels. For example, is TFAM a substrate of CLPX? What is the timing of TFAM disappearance on mtDNA relative to CLPX recruitment? The authors could address this by staining for TFAM in each of the mutant/overexpression conditions. The authors mention in the second paragraph of the introduction that in humans, TFAM re-localizes to the nucleus, rather than being degraded. Is this relocalization conserved in flies? If so, this has implications for the model that Poldip2 recruits CLPX to degrade TFAM. The manuscript would benefit from additional discussion clarifying these possibilities.

We would like to thank all the reviewers for the time to read our work and provide constructive comments. During our revision time, we performed a number of experiments and added new data along with related descriptions and discussions to the revised manuscript. We believe the new data and writing have improved the manuscript and further strengthened the connect between POLDIP2 and mtDNA elimination in *Drosophila* spermatids. Below, we provide detailed replies to individual comments.

Referee #1:

The study by Wang et al. used an unbiased genetic screen to identify new genes required for mtDNA elimination during spermatogenesis. Wang et al. focus on Poldip2 (also identified by Chen et al.) and show that is required for complete paternal mtDNA elimination. Similar to Chen et al., Wang et al. show that Poldip2 mutants are semi sterile. However, Wang et al. additionally show that Poldip2 mutant fathers can pass their mtDNA to many (or perhaps all) of their offspring. Why mtDNA is almost always inherited from a single parent is an unresolved and broadly interesting question in genetics, that will be interesting to the readers of EMBO. While Wang et al. do not resolve this question, their discovery that Poldip2 fathers can transmit mtDNA to their offspring is a very important first step that will promote future research on the topic. The following discoveries in Wang et al. are important and are well supported by the data:

- 1) Poldip2 is required for complete elimination of mtDNA during late spermatogenesis
- 2) Poldip2 is a mitochondrial matrix protein that can associate with mtDNA
- 3) Poldip2 mutants are semi sterile
- 4) Poldip2 mutant fathers transmit mtDNA to their offspring
- 5) Poldip2 physically interacts with ClpX
- 6) Similar to Poldip2, ClpX is required for complete mtDNA elimination during late spermatogenesis.

There are no major problems with this paper and in my opinion, it is suitable for publication in your journal with only minor revisions.

Minor Criticisms:

1) The authors' EMS screen seems to be a highly successful method for studying mtDNA elimination during spermatogenesis. The authors found 47 mutant lines with high mtDNA retention but focused on one line, #23, to characterize in this paper. Do the other lines harbor alleles of Poldip2, or the other two genes previously reported to be involved in mtDNA elimination during spermatogenesis: EndoG and Tamas? The strains that the authors mutagenized should also allow them to roughly map the causative allele to one of 4 chromosome arms. Did the authors do this mapping? If so, did they perform complementation tests with mutants that mapped to the same arm? From their screen, can the authors estimate how many genes contribute to mtDNA elimination during spermatogenesis? Or can they at least comment on how many new genes they found?

We apologize for not adding more information about our EMS screen in our initial submission. We generated EMS-mutagenized fly lines by crossing EMS-treated adult males with females carrying 2nd or 3rd chromosome balancers. This approach allowed us to determine which chromosome carries the mutations for each line, but not the chromosome arms, as we focused on homozygous viable mutants. Of the 47 positive lines, 14 have mutations on the 2nd chromosome and 33 on the 3rd. We conducted a complementation test on the 3rd chromosome EMS lines and identified four complementary groups. Based on this, we estimate that >40 genes are involved in the elimination process. Notably, we did not identify EndoG or Tamas, suggesting that the screen is not saturated. This may be due, in part, to the fact that we only screened homozygous viable mutants. We have included this information in the revised

manuscript. We further acknowledged in the discussion that in POLDIP2 KO mutants, the mtDNA copy number in late spermatids is still significantly reduced compared to spermatids in the earlier elongation stage. This suggests that other players regulating mtDNA elimination in *Drosophila* sperm are yet to be identified.

2) The paragraph about the mt:ND2del1 assay to measure paternal leakage doesn't refer to any data presented as figures and should be removed.

We consider this paragraph is essential to help readers understand the setup of our paternal leakage test, as it introduces and explains the rationale for selecting the D.mel and D.yak combination to detect paternal leakage (used in the subsequent paragraph and Figure 2F). It also highlights that the level of leakage is low. Hence, we have decided to keep it.

3) The Dmel x Dyak assay to detect paternal leakage is very elegant. To strengthen their conclusion that Poldip2 but not WT fathers transmit mtDNA to their adult offspring, the authors should report the raw data for the control WT cross- particularly because Chen et al. did not detect paternal transmission. The authors should also consider reporting the sensitivity threshold of their assay. For example, could their assay detect 0.0001% Dmel in Dyak? From the measured sensitivity, do they think they could detect a single mtDNA molecule inherited through a sperm?

In the revised version, we have included the raw ddPCR data presented in Fig 2 as source data. We also have added ddPCR plots showing various titrations (0.5%, 0.05%, 0.005%, and 0.0005%) of paternal mtDNA to demonstrate the sensitivity of our assay (new Fig S2F). The data indicate that the detection limit is between 0.005% and 0.0005%. Since each adult male has approximately 10^8 copies of mtDNA, our assay is unable to detect a single molecule inherited through sperm, which would be on the order of 10^{-8} .

4) Similarly, the authors state, "we anticipate the initial leaked amount in embryos to be much lower than the ddPCR values." The authors should expand on this interpretation to help the reader follow. How many copies of mtDNA are in an egg and what is the range of copy numbers (and paternal mtDNA percentages) that could be transmitted by sperm? If a sperm has at most 200 copies and an egg has about 2 million copies, one might observe initial paternal transmission to be anywhere from 0 to 0.01%.

We thank the reviewer for this suggestion. The *Drosophila* egg contains approximately 10^7 copies of mtDNA. During the revision time, we estimated the average amount of mtDNA retained in mature sperm of POLDIP2^{-/-} by quantifying the total number of dsDNA puncta in the seminal vesicle and dividing it by the number of sperm heads (Z-stack confocal imaging). The estimated count is around 6.6 copies per sperm (Fig 2G in the revised manuscript). This may be an underestimation, as our immunostaining efficiency is not 100% and each dsDNA signal could represent more than one mtDNA copy. Nonetheless, this number suggests that the initial proportion of paternal mtDNA would be ~0.0001%, which falls below our ddPCR detection limits.

In most of our crosses, paternal mtDNA was detected at around 0.01% in 2-day-old adults, indicating a 100-fold increase from the embryonic to the early adult stage. In earlier work, we demonstrated that *D. melanogaster* mtDNA could outcompete *D. yakuba* mtDNA within two generations (Ma et al., 2016). Therefore, the significant increase in paternal mtDNA copy number from the embryonic to adult stage is not unexpected. We have included this new data and related discussion in the revised manuscript as suggested by the reviewer.

5) Regarding the TFAM ChIP qPCR experiments, the authors state "This significantly higher mtDNA enrichment compared to POLDIP2 is anticipated, as TFAM is known to bind broadly across the mitochondrial genome with high affinity and quantities. The difference in mtDNA enrichment between TFAM and POLDIP2 ChIP assay likely arises from differences in their mtDNA binding affinities and quantities." An alternative explanation is that different antibodies have different affinities for target proteins and sources of background. The authors should just remove the above statement and the comparison between Poldip2 and TFAM binding to mtDNA because it is not essential or important for the paper.

This is a good point. We have revised our description for this part.

6) In the next paragraph, the authors state, "The observed enrichment of mtDNA with POLDIP2 immunoprecipitation is unlikely attributed to interactions between POLDIP2 and other mtDNA binding proteins, as our co-immunoprecipitation (co-IP) experiments detected weak/no interaction between POLDIP2 and TFAM, mtSSB, or mtDNA polymerase POLG1 in fly tissues and HEK293 cells (Fig 3G, S3D)." These co-IP experiments have much less sensitivity than ChIP. Failing to detect a protein-protein interaction in co-IP does not mean that this interaction is not part of a protein-protein-DNA interaction in ChIP. The authors should just remove the above statement.

We agree with the reviewer and have removed this statement. Moreover, we added new co-IP data that examined the interaction between POLDIP2 and TFAM or POLG1 with and without crosslinking reagents. This data showed that POLDIP2 interacts with TFAM and POLG1 only when samples were crosslinked, suggesting that while POLDIP2 is unlikely to form a stable interaction with TFAM or POLG1, it may localise near mtDNA nucleoids and transiently associate with these proteins (Fig 3F).

7) Chen et al. found that Poldip2 expression coincided with mtDNA elimination during late spermatogenesis. An interesting question is whether Poldip2 expression induces mtDNA elimination. The authors created ubi-Poldip2 transgenic fly lines, which theoretically constitutively express Poldip2. Did these lines constitutively express Poldip2 and did this result in reduced mtDNA in any of the tissues expressing Poldip2?

We measured the total mtDNA copy number in the soma of adult males overexpressing poldip2-PA and PB and found that they did not show reduced mtDNA numbers. This suggests that POLDIP2 is not sufficient to induce mtDNA degradation outside the testis. We have included this data (Fig S5E) and related discussion in the revised manuscript.

8) The connection between Poldip2 and ClpX is interesting because both genes physically interact and have a similar mtDNA elimination defect. However, the author's model where Poldip2 recruits ClpX to mtDNA to help remove mtDNA-interacting proteins is quite speculative and belongs in the discussion section.

We agree with the reviewer that this model is speculative. In the revised version, we provided new immunoblots showing that the TFAM level was doubled in the *Poldip2* and *ClpX* double mutant, while the POLG1 levels remained comparable to that of the wild type (Fig 4H). We also performed co-IP and showed that CLPX-TFAM interaction was detected when samples were pre-treated with crosslinking reagents (Fig S5C). These new data raise the possibility that TFAM might be a substrate of CLPXP. We have included new data and extra descriptions in the discussion section.

9) In the discussion section, the authors speculate, "POLDIP2 could undergo continuous degradation via post-translational regulation in earlier stages to prevent it from interacting with mtDNA and trigger its degradation." The authors should be able to test this by looking at their ubi-Poldip2 transgene, which is presumably constitutively expressed.

This is a valid point. We did not see a reduced expression of POLDIP2 in sperm cells before the spermatid stage of Ubi-POLDIP2 flies. However, this continuous presence of POLDIP2 could be an artefact of its high overexpression in the Ubi-Poldip2 lines, which could mask the degradation process that would occur under normal physiological conditions. Another possibility is that high levels of Ubi-Poldip2 mRNA might be influencing this, as the UTRs of Poldip2, which could affect the mRNA stability of endogenous Poldip2, were not included when generating the Ubi-Poldip2 flies. Considering these potential issues with our transgenic lines, we have decided not to address this point in our discussion.

Referee #2:

Using forward genetic screening, Wang et al. identified POLDIP2 as a factor regulating mtDNA clearance during spermatogenesis in *Drosophila melanogaster*. They showed that POLDIP2 is a mitochondrial matrix protein and binds to mtDNA in ChIP assay. Using mass spec analysis, they further identified CLPX, a subunit of matrix protease complex CLPXP, as a POLDIP2 binding protein. The ClpX hypomorphic mutant also showed a defect in mtDNA clearance in sperm, and this phenotype was rescued by Poldip2 overexpression. These results suggest that POLDIP2 and CLPX function together in mtDNA clearance. Overall, this work is well organized and provides important mechanistic insights into mtDNA clearance in fly sperm. Although many of the conclusions are well supported by the data, some important points should be addressed before publication.

Major comments

1. In Fig.2, the authors showed that the mtDNA signals remain in the seminal vesicles of the Poldip2 mutant. It would be better to include some quantification of remaining mtDNA nucleoids in each sperm.

During the revision, we used Z-stack confocal imaging to quantify the total number of dsDNA puncta in the seminal vesicle. This number was then divided by the number of sperm heads to estimate the average mtDNA copies retained by individual mature sperm of POLDIP2^{-/-}. The estimated count is around 6.6 copies (Fig 3G). This may be an underestimation, as our immunostaining efficiency is not 100% and each dsDNA signal could represent more than one mtDNA copy. We have included the new data and related discussion in the revised manuscript.

2. We cannot see whether the organization of mtDNA nucleoids is affected by the Poldip2 mutation in Fig2. Please mention this point (size, number, and position, for example). It would also be informative to include enlarged images of the wild-type and mutant spermatids at various stages to compare morphology and number of nucleoids.

Our confocal imaging, with a resolution of 250nm, is insufficient to capture detailed organisational changes in mtDNA nucleoids and their morphology, which are approximately 400nm in diameter (including amplification from primary and secondary antibodies). As a result, a zoomed-in view of our dsDNA foci would not provide substantial information. Nevertheless, we quantified the nucleoid size to investigate whether there are any obvious changes in nucleoid packing in the Poldip2 KO mutant. Based on our measurements across various stages of spermatogenesis, we observed no significant differences

in nucleoid diameter between the wild type and the mutant. We have included these new findings (Fig S4B) and corresponding descriptions in the revised manuscript.

3. The authors showed that mCherry-tagged POLDIP2 PA and PB colocalized with mitochondrial markers. It should be mentioned whether these tagged transgenes are functional. I also wonder whether POLDIP2 localizes to mtDNA nucleoids or localizes uniformly in the matrix. Enlarged images of Fig. 3C and D would be helpful to address this point.

Our rescue experiments indicate that transgenic POLDIP2 proteins are functional (Fig 2D, E, Fig 4E, Fig S4F).

Regarding POLDIP2 localisation in the mitochondrial matrix, we attempted to examine the co-localisation of dsDNA foci and POLDIP2-mCherry signals by confocal microscopy. However, the high level of POLDIP2 expression in the Ubi-Poldip2 transgenic lines and the limited resolution of our confocal imaging made it challenging to assess the co-localisation. The POLDIP2-mCherry signals appeared diffused overall, although some enrichment at dsDNA foci was observed at the spermatocyte stage (Fig S3C). This could be due to overexpression saturating the mitochondrial matrix. Alternatively, only a small number of POLDIP2 molecules could bind to each mtDNA nucleoid, resulting in an unclear puncta signal. Overall, our confocal imaging lacks the resolution to distinguish subdomains within the mitochondria. In the revised manuscript, we provided enlarged images (Fig S3C) to show the POLDIP2 mCherry localisation related to dsDNA foci.

Moreover, we performed new co-IP experiments, which detected interactions between POLDIP2 and TFAM, and POLDIP2 and mtDNA polymerase POLG1, when cross-linking was applied during sample preparation (Fig 3F). This suggests that POLDIP2 is unlikely to form a stable interaction with TFAM or POLG1 but might localise near mtDNA nucleoids in the mitochondrial matrix. This new piece of data, alongside our ChIP-qPCR, demonstrates that POLDIP2 can localise close to mtDNA.

4. Is the amount of TFAM or other nucleoid proteins (e. g. mtSSB, POLG1) in the testis affected by the Poldip2 and ClpX mutations? It would be also important to examine the protein dynamics of ClpX during spermatogenesis. Does ClpX localize to nucleoid in this process?

We measured the amount of TFAM and POLG1 in the Poldip2 and ClpX double mutant, and found that the TFAM level was doubled, whereas the POLG1 level remained similar to that of the wild type (Fig 4H). We could not estimate endogenous mtSSB levels by Western Blot, as our mtSSB-GFP line is an overexpression line under the ubi promoter, and the available antibodies against human mtSSB do not recognise the *Drosophila* mtSSB.

We also examined ClpX dynamics during spermatogenesis by immunostaining (Fig S5B). Our data showed that the ClpX level remained relatively high throughout spermatogenesis, until very late spermatid stage where DJ-GFP expression became prominent - a marker for fully elongated spermatid bundles or individualised sperm (Santel et al, 1997). This divergence in timing suggests that CLPX has broader roles in mitochondrial function beyond mtDNA elimination. Our confocal imaging does not indicate a clear co-localisation of ClpX and nucleoid in spermatids, as the expression level of ClpX is high. However, we performed co-IP, which detected interaction between TFAM and CLPX in the testis when samples were crosslinked (Fig S5C). This new piece of data indicates that CLPX can localise close to mtDNA. We have added all the new data and related discussions in the revised version.

5. The Poldip2 mutant males showed reduced fertility. Please clarify the morphology and number of mature sperm in the Poldip2 mutant.

It is not easy to count the number of mature sperm as they are continuously made, and sperm stored in seminal vesicles are continuously emptied through mating. Hence, the fertility assay has been used commonly in the fly community to reflect the quality and quantity of sperm. Our confocal imaging also did not detect obvious defects with *Poldip2* KO sperm.

6. In Fig. 4D, the authors showed that the phenotype of the *ClpX* mutant was rescued by *Poldip2* overexpression by using qPCR of mtDNA in the testis. Because the testis also includes somatic cells, it is better to confirm this result by mtDNA staining of sperm in seminal vesicles.

We have added an image of a seminal vesicle isolated from *ubi-Poldip2-PB;ClpX^{LA00797}* flies as Fig S4F in the revised manuscript.

7. *Poldip2* was originally identified as a human polymerase δ P50-interacting protein and has been proposed to be involved in nuclear genome replication and repair. In this study, the authors seem to agree with the conclusion proposed by Chen et al that *Poldip2* functions as a DNA exonuclease. Does *Poldip2* contain any conserved DNA exonuclease catalytic domain? If not, the authors should confirm the DNA exonuclease activity of *Poldip2* in vitro by themselves. It would be essential to examine DNA-binding defective mutant (Yvvc domain mutations) and hypomorphic mutants identified by the authors as well as wild-type proteins.

We appreciate the suggestion to investigate this further, but it falls outside the focus of the current research. We neither agree nor disagree with the conclusions of Chen et al. regarding *POLDIP2*'s exonuclease activity, as we do not have data to confirm or refute their findings. Our intention in citing their work was to acknowledge the possibility of such a function for *POLDIP2*, but we do not provide experimental evidence in support of this in our study.

8. It has been reported that *rab7* and *atg7* are involved in the degradation of paternal mitochondria in embryos. Do these mutations (*rab7* or *atg7*) enhance the paternal mtDNA leakage phenotype of *Poldip2* and *ClpX* mutants? Do *Poldip2* and *ClpX* double mutants show enhanced paternal mtDNA leakage phenotype?

This is an interesting point. We did not test the paternal leakage of the *Poldip2* and *ClpX* double mutant because of their reduced male fertility. However, we overexpressed a dominant negative *Rab7* in females, which was previously shown to delay the degradation of paternal mtDNA after fertilization (Politi et al 2014). We found that progeny from crosses using *POLDIP2*^{-/-} males with females expressing the dominant negative *rab7*, leads to a slight increase in paternal leakage levels. However, due to large variations between individual crosses within each group, the increase is not statistically significant (Fig S5D). Hence, it remains uncertain whether the delayed paternal mitochondrial degradation in embryos could enhance paternal leakage. We have included the new data and related discussion in the revised manuscript.

Minor comments

1. It would be informative to mention the expression pattern of *CLPX* and the progeny size of the mutant. We have provided immunostaining images of *CLPX* at different stages of spermatogenesis in the revised manuscript (Fig S5B). As mentioned above, the *CLPX* level remained high throughout spermatogenesis, until very late-stage spermatids where DJ-GFP expression became prominent. Regarding the progeny size, we attempted to establish an isogenic line of *ClpX^{LA00797}* line in the w1118 background to perform fertility assays, but failed to do so within the timeframe of this revision. Given this information is not essential for any of the conclusions in our paper, we decided not to pursue it further.

2. Fig. 1C, Fig. 3E-G, Fig. 4A-C, F, Fig. S4B, C

Please indicate which tissue was used in the legend of each Figure (or directly in the figures).

We have added the information about tissues used in the figure legends.

3. In Fig. 2F, paternal leakage of mtDNA was addressed by ddPCR. Please indicate which stage of progeny was used in the main text and Figure 2 legend (According to Methods, adult progeny was used).

2 day old adult progeny were used. We have added the information in the main text, Fig 2 and figure legend in the revised manuscript.

4. Please indicate the position of P-element insertion for Poldip2 EY08866 and CLPX LA00797.

We have presented the information in the revised Fig S2C, and Fig S4D.

5. p6 line 2

"On average, each male adult carries $\sim 10^8$ copies of mtDNA."

We have revised our writing and added the total number of mtDNA in embryos and the citation.

6. Fig. S2D right panel

The genotype is indicated as ubi-Poldip2-PB-FLAG-mCherry; Poldip2KO in the legend but ubi-Poldip2-PA; Poldip2KO in Figure.

We have checked the labels to ensure the consistency between the figure and the legend.

Referee #3:

Summary and significance:

Wang et al. report the finding of Poldip2 through a forward mutagenesis screen in *Drosophila melanogaster*, identifying 47 mutants that exhibited retention of mtDNA in mature sperm. Using deficiency mapping and sequencing, the authors mapped one of the mutants to the Poldip2. The authors use proteinase K protection assays, ChIP, and fluorescence imaging to characterize Poldip2 as a mitochondrial matrix protein that can bind to mtDNA. Poldip2 mutant males are able to transmit their mtDNA to their offspring if mated with *D. melanogaster* females carrying exclusively *D. yakuba* mtDNA, which is less competitive in the background of a *D. melanogaster* nuclear genome. They show that Poldip2 expression coincides with mtDNA degradation in the testis, and show that Poldip2 interacts and stabilizes CLPX, a protease. A CLPX hypomorphic allele that leads to a substantial reduction in protein levels phenocopies the Poldip2 mutant, which can be rescued by overexpression of a Poldip2 isoform. The authors suggest that CLPX contributes to mtDNA clearance in testes, presumably in a co-regulative process together with Poldip2.

We consider this study an interesting and important body of work. The experiments are well-designed, and the data is presented clearly. The identification of factors contributing to mtDNA clearance in *Drosophila* sperm adds an important piece to the puzzle of safeguarding maternal inheritance of mitochondrial DNA.

Specific concerns:

The manuscript would benefit from a clearer description of the nature of the interaction between Poldip2 and CLPX as well as the potential function of CLPX during this process. The proposed model in Fig 4

hints towards a possible influence on TFAM levels bound to mtDNA, but this is speculative as there is no data in the paper that speaks specifically to the impact on TFAM levels. For example, is TFAM a substrate of CLPX? What is the timing of TFAM disappearance on mtDNA relative to CLPX recruitment? The authors could address this by staining for TFAM in each of the mutant/overexpression conditions. The authors mention in the second paragraph of the introduction that in humans, TFAM re-localizes to the nucleus, rather than being degraded. Is this relocalization conserved in flies? If so, this has implications for the model that Poldip2 recruits CLPX to degrade TFAM. The manuscript would benefit from additional discussion clarifying these possibilities.

This is a good suggestion. During our revision, we performed TFAM staining and showed that it does not translocate from mitochondria to the nucleus in *Drosophila* spermatids or mature sperm (Fig S5F). Hence, detailed mechanisms for eliminating paternal mtDNA during spermatogenesis appear to differ between *Drosophila* and humans. On the other hand, we found many dsDNA loci in mature sperm of Poldip2^{-/-} mutant co-stained with TFAM, indicating that retained mtDNA molecules were still associated with TFAM (Fig 4I). This suggests that POLDIP2 might be required to remove TFAM from mtDNA nucleoid in the late spermatid stage.

Quantifying TFAM levels during spermatogenesis via imaging is challenging, as it is hard to stage spermatid bundles from different testes. Such information is essential to compare TFAM signals at different stages of elongation in the wild type and poldip2 mutant. The immunostaining signal is also not quantitative enough for us to draw clear conclusions. Instead, we examined TFAM levels by Western Blot. Our immunoblots showed that the TFAM level was doubled in the Poldip2 and ClpX double mutant, whereas the POLG1 level remained comparable to the wild type (Fig 4H). We further performed co-IP and showed that CLPX can interact with TFAM when the samples were crosslinked (Fig S5C). These new observations raise the possibility that TFAM could be a substrate of CLPXP. We also examined the CLPXP expression pattern by immunostaining (Fig S5B), which showed that the CLPXP level remained high throughout spermatogenesis, until very late-stage spermatids where DJ-GFP expression became prominent. Further investigations with more temporal and spatial resolution are required to examine the dynamics between CLPXP and TFAM interaction at different stages of spermatogenesis, and whether POLDIP2 mediates such an interaction. We have added all the new data and related discussion in the revised manuscript.

Dear Hansong,

Thank you for submitting a revised version of your manuscript. We have now received input from two of the original reviewers, who find that most of previous concerns have been addressed satisfactorily and recommend acceptance of the manuscript after a minor textual revision as requested by reviewer #2.

Additionally, there remain a few editorial points that need addressing before I can extend official acceptance of the manuscript:

1. Please reduce the number of keywords to five.
2. To appropriately register the funding information in our online system, please remove from the Comments box and inserted as separate entries the following funding sources - The Gurdon Institute Core Facility is funded by Wellcome Trust grant 203144 and CRUK grant C6946/A24843.
3. Please make sure that the order of the sections in the manuscript is as follows: Abstract / Keywords / Introduction / Results / Discussion / Methods / Acknowledgments / Disclosure and competing interests statement / References / Figure legends / Tables and their legends / Expanded View Figure legends.
4. All Materials and Methods need to be described in the main text using our 'Structured Methods' format. According to this format, the Methods section includes a Reagents and Tools Table (listing key reagents, experimental models, software and relevant equipment and including their sources and relevant identifiers) followed by a Methods and Protocols section describing the methods, ideally using a step-by-step protocol format. The aim is to facilitate adoption of the methodologies across labs. Please download and fill our Reagents and Tools Table template (.docx), which you can find in our author guidelines: <https://www.embopress.org/page/journal/14602075/authorguide#structuredmethods>
When submitting your revised manuscript, please do not include the Reagents and Tools Table in the Methods section of the manuscript but upload it as a separate file choosing the file type "Reagent Table".
An example of a Method paper with Structured Methods can be found here: <https://www.embopress.org/doi/10.15252/msb.20178071>.
5. Please rename "Data and materials availability" section into "Data availability" and provide resolvable URLs to the deposited sequencing datasets.
6. Please rename "Competing interests" section into "Disclosure and competing interests statement" (further info: <https://www.embopress.org/page/journal/14602075/authorguide#conflictsofinterest>).
7. CRediT has replaced the traditional author contributions section because it offers a systematic, machine-readable author contributions format that allows for more effective research assessment. Please remove the Authors Contributions from the manuscript and use the free text boxes beneath each contributing author's name in our online submission system to add specific details on the author's contribution. More information is available in our guide to authors.
8. Please submit Appendix file in the pdf format. Please update the nomenclature to Appendix Table S1-S3 and update the callouts accordingly throughout the manuscript.
9. Please rename the title title "Supplementary figure legends" in the manuscript text file to "Expanded View Figure Legends" and update the nomenclature to that of Figure EV1 etc. throughout the manuscript.
10. Our data editors have flagged the following issues in figure legends that need correcting:
 - Please define the annotated p values ****/****/**/* in the legend of figure 2E, 3B, 4E as appropriate.
 - Please provide the exact p values in the legend of figure 3B.
 - Please note that in figure 2G there is a mismatch between the annotated p values in the figure legend and the annotated p values in the figure file that should be corrected.
11. Papers published in The EMBO Journal are accompanied online by a 'Synopsis' to enhance discoverability of the manuscript. It consists of A) a short (1-2 sentences) summary of the findings and their significance, B) 3-4 bullet points highlighting key results and C) a synopsis image that is 550x300-600 pixels large (width x height, jpeg or png format). You can either show a model or key data in the synopsis image. Please note that the image size is rather small and that text needs to be readable at the final size. Please send us this information together with the revised manuscript.

With best wishes,

Ieva

Ieva Gailite, PhD
Senior Scientific Editor
The EMBO Journal

Meyerhofstrasse 1
D-69117 Heidelberg
Tel: +4962218891309
i.gailite@embojournal.org

We realize that it is difficult to revise to a specific deadline. In the interest of protecting the conceptual advance provided by the work, we recommend a revision within 3 months (6th Mar 2025). Please discuss the revision progress ahead of this time with the editor if you require more time to complete the revisions.

Referee #2:

The authors have addressed most of the concerns of the reviewers, and the manuscript has been substantially improved. Please just confirm the following minor point.

1. Fig.4A

Please indicate which tissue was used for sample preparation in the figure legend.

Referee #3:

We read the revised manuscript and the response to the reviewers and found that the authors addressed all of the reviewers' concerns. This is an interesting manuscript, and we support publication.

The authors addressed the remaining editorial issues.

Dear Hansong,

Thank you for submitting the final revised version and addressing the remaining editorial points. I sincerely apologise for the delay in communicating the decision due to the coordination of article processing on our side. I am now pleased to inform you that your manuscript has been accepted for publication. Congratulations on a great study!

Before we forward your manuscript to our publishers, I would like to propose some edits in the manuscript title, abstract and synopsis (please also see the attached file). I have also written a short blurb that will accompany the title of your manuscript in our online system. In particular, I have adjusted the nomenclature to that used in *Drosophila* in the manuscript title and abstract. Please take a look and let me know if any corrections are needed.

Title:

Poldip2 promotes mitochondrial DNA elimination during *Drosophila* spermatogenesis to ensure maternal inheritance

Blurb:

The mitochondrial matrix protein Poldip2 interacts with mtDNA and the mitochondrial protease subunit ClpX to mediate paternal mtDNA elimination.

Synopsis:

In many metazoans, including *Drosophila* and humans, paternal mitochondrial DNA (mtDNA) is eliminated during spermatogenesis to ensure maternal mtDNA inheritance. This study uncovers the role of Poldip2 in regulating this process during late *Drosophila* sperm development.

- A genetic screen identifies mutations that cause retention of paternal mtDNA in *Drosophila* mature sperm.
- Disrupting the Poldip2 gene leads to high levels of paternal mtDNA retention in mature sperm and low levels of paternal mtDNA leakage to offspring.
- Poldip2 is a mitochondrial matrix protein capable of binding mtDNA.
- The mitochondrial protease component ClpX interacts with Poldip2 and likely co-regulates mtDNA elimination in *Drosophila* spermatids.

If you have any questions, please do not hesitate to contact the Editorial Office. Thank you for this contribution to The EMBO Journal and congratulations on a nice study!

With best wishes,

leva

leva Gailite, PhD
Senior Scientific Editor
The EMBO Journal
Meyerohofstrasse 1
D-69117 Heidelberg
Tel: +4962218891309
i.gailite@embojournal.org

>>> Please note that it is The EMBO Journal policy for the transcript of the editorial process (containing referee reports and your

response letter) to be published as an online supplement to each paper. If you do NOT want this, you will need to inform the Editorial Office via email immediately. More information is available here: https://www.embopress.org/transparent-process#Review_Process